# DIVERSITY-REWARDED CFG DISTILLATION

**Geoffrey Cideron[†], Andrea Agostinelli, Johan Ferret, Sertan Girgin, Romuald Elie
Olivier Bachem, Sarah Perrin*, Alexandre Ramé***

Google DeepMind, * Equal advisory contribution

## ABSTRACT

Generative models are transforming creative domains such as music generation, with inference-time strategies like Classifier-Free Guidance (CFG) playing a crucial role. However, CFG doubles inference cost while limiting originality and diversity across generated contents. In this paper, we introduce diversity-rewarded CFG distillation, a novel finetuning procedure that distills the strengths of CFG while addressing its limitations. Our approach optimises two training objectives: (1) a distillation objective, encouraging the model alone (without CFG) to imitate the CFG-augmented predictions, and (2) an RL objective with a diversity reward, promoting the generation of diverse outputs for a given prompt. By finetuning, we learn model weights with the ability to generate high-quality and diverse outputs, without any inference overhead. This also unlocks the potential of weight-based model merging strategies: by interpolating between the weights of two models (the first focusing on quality, the second on diversity), we can control the quality-diversity trade-off at deployment time, and even further boost performance. We conduct extensive experiments on the `MusicLM` text-to-music generative model, where our approach surpasses CFG in terms of quality-diversity Pareto optimality. According to human evaluators, our finetuned-then-merged model generates samples with higher quality-diversity than the base model augmented with CFG. Explore our generations at google-research.github.io/seanet/musiclm/diverse_music.

## 1 INTRODUCTION

**Generative models for creative domains.** Art and entertainment — domains historically driven by human creativity — are undergoing a profound transformation thanks to AI generative models. These models, often powered by Large Language Models (LLMs) or diffusion, can now generate texts (Gemini Team, 2023), images (Ramesh et al., 2022), videos (Ho et al., 2022), and audios (Borsos et al., 2023; Kreuk et al., 2023; Copet et al., 2023; Agostinelli et al., 2023; Cideron et al., 2024; Défossez et al.). To further refine the quality, these models are often augmented with inference methods during real-world deployment, ranging from simple temperature scaling to more refined methods like Beam search (Freitag & Al-Onaizan, 2017), test-time augmentation (Shanmugam et al., 2021), or MCTS (Kocsis & Szepesvári, 2006). A particularly popular method for image and audio generation is classifier-free guidance (CFG) (Ho & Salimans, 2022), *e.g.* used in DALL-E (Ramesh et al., 2022) or AudioGen (Kreuk et al., 2023). CFG improves the model's fidelity to the prompt by combining the logits of conditional and unconditional generations. Despite its benefits, CFG has two main limitations: it doubles the computational cost during deployment and reduces the diversity of generated content (Ho & Salimans, 2022; Dhariwal & Nichol, 2021; Kreuk et al., 2023; Meng et al., 2023), hindering the exploration of novel and diverse ideas — a cornerstone of creativity. Ideally, these models should not only fulfill user intent but also surprise them with unexpected and innovative outputs: the model should not generate systematically the same content (Hamilton, 2024).

**Quality-diversity trade-off.** Effectively controlling the quality-diversity trade-off is thus extremely important but challenging. On the one hand, optimising quality usually reduces diversity; this limitation affects inference methods such as CFG but also finetuning stages such as RLHF (Kirk et al., 2024; Mohammadi, 2024). On the other hand, promoting diversity usually reduces quality (Brown

---

[†]Correspondence to: Geoffrey Cideron <gcideron@google.com>

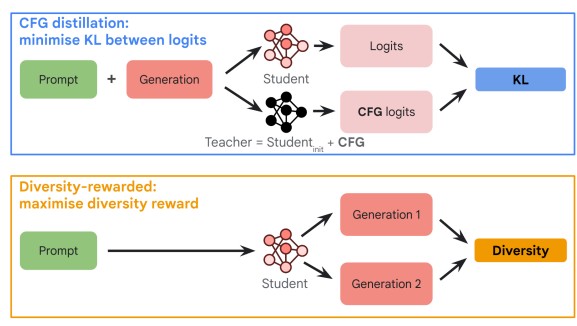 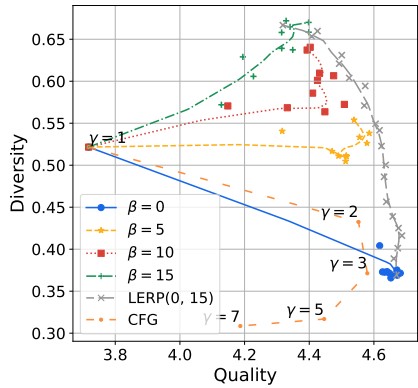

Figure 1: **Left.** Illustration of the two objectives: CFG distillation (above) and the diversity reward (below), multiplied by the diversity coefficient $\beta$ in the joint finetuning objective. **Right.** Quality-diversity trade-off for different strategies. The first four lines represent the training trajectories of our approach, distilling CFG (with $\gamma = 3$) with varying diversity coefficient $\beta$ in $\{0, 5, 10, 15\}$; every 500 training steps, we evaluate the quality and diversity of the generations. Larger values of $\beta$ lead to more diverse models yet slightly less quality. For linear interpolation (LERP), each cross corresponds to a $0 \leq \lambda \leq 1$ when interpolating between the weights $\theta_q$ of a quality-focused model ($\beta = 0$) and those $\theta_d$ of a diversity-focused model ($\beta = 15$); the evaluated generations are obtained from the weights $(1 - \lambda) \cdot \theta_q + \lambda \cdot \theta_d$. For the CFG baseline, each dot corresponds to a different value for the guidance factor $1 \leq \gamma \leq 7$. This plot shows that our method improves the quality-diversity trade-off; notably, LERP uncovers a strong and steerable front of solutions by just interpolating between the weights of two models, at deployment time.

et al., 2005), *e.g.* when increasing temperature at inference (Zhang et al., 2021). As further discussed in Section 4, quality-diversity algorithms (Lehman & Stanley, 2011; Mouret & Clune, 2015) seek to train a population of models with diverse abilities. In contrast, optimising directly the diversity of generations of a single model is an under-explored yet promising avenue, that we investigate here.

**Diversity-rewarded CFG distillation.** In this work, we introduce a novel finetuning strategy to enhance the quality-diversity trade-off in generative models for creative domains. Specifically, we combine distillation and reinforcement learning (RL) to optimise two complementary objectives. The first is a novel *CFG distillation objective* for LLMs where we distill the behavior of a teacher into the student. Critically, the teacher is the CFG-augmented base model, rather than a larger third-party model as commonly done in the literature (Hinton et al., 2015). This involves minimizing the KL divergence between the logits of the teacher and the student on the data distribution generated by the student (to reduce train-test mismatch), following the on-policy distillation framework of Agarwal et al. (2024). By distilling CFG into the model weights, we improve generation quality while eliminating CFG's inference overhead. The second is a novel *RL with a diversity reward* objective, maximising the diversity across pairs of generations for a given prompt. Diversity is measured with by comparing pairs of generations by first embedding them and then computing their negative (cosine) similarity in this embedding space. Combining these two objectives allows the finetuned model to inherit the quality of CFG without its cost, while simultaneously maintaining diversity through the RL objective, thus solving the two main drawbacks of CFG at inference time.

**Model merging.** The hyperparameter $\beta$ (multiplier for the diversity reward) allows controlling the quality-diversity trade-off *at training time*, as shown in Figure 1 (right); in contrast, traditional CFG can do it *at deployment time*. To enable this level of control within our approach, we propose a third contribution involving model merging. Specifically, we finetune two models, one focusing on quality (low $\beta$) and the other focusing on diversity (high $\beta$), and then combine their weights by *linear interpolation (LERP)* (Utans, 1996), balancing between quality and diversity. This follows from the linear mode connectivity property (Frankle et al., 2020) and recent advances in model merging (Wortsman et al., 2022a; Ramé et al., 2022; Ilharco et al., 2023), showing that weights finetuned from a shared pretrained initialisation can be interpolated, despite the non-linearities in the architecture.

Notably, Ramé et al. (2023) showed that interpolating between weights finetuned on different rewards trades their abilities off. Thus, we interpolate between the weights of a quality-focused model and a diversity-focused model: sliding the interpolating coefficient $\lambda$ between $0$ and $1$ uncovers a strong and steerable quality-diversity front of solutions, without overhead at deployment. Crucially, we find that the interpolated model using $\lambda = 0.5$ is our best model, outperforming models explicitly finetuned for intermediate values of the diversity hyperparameter $\beta$.

> **Contribution 1: *CFG distillation for quality in LLMs*.** We distill the quality benefits of a costly inference-time strategy, CFG, into the weights of our model.

> **Contribution 2: *Reinforcement learning for diversity*.** We reward the model to create diverse generations, reducing the drop in diversity caused by CFG and finetuning.

> **Contribution 3: *Model merging for Pareto-optimality*.** We trade-off quality and diversity at deployment time by interpolating between the weights of a quality-focused model and a diversity-focused model.

**Music generation.** We apply our strategy to text-to-music generation, a creative task where balancing quality and diversity is key. Specifically, we finetune MusicLM (Agostinelli et al., 2023) and consistently improve the quality-diversity trade-off achieved by the CFG-augmented previous state-of-the-art (Cideron et al., 2024). Our experiments, featuring human evaluations, validate that our models generate more diverse music while maintaining high quality.

## 2 DIVERSITY-REWARDED CFG DISTILLATION

**Notations.** Let $m_\theta$ denote an auto-regressive model parameterised by $\theta$. Given an input sequence $x$ from the space of sequences $X$, the model $m_\theta$ defines a policy $\pi_\theta$ by sequential sampling of an output sequence $y = (s_1, ..., s_L)$ of length $L$. Specifically, given a partial sequence $y_{<n} = (s_1, ..., s_{n-1})$, the policy $\pi_\theta$ samples the next token $s_n$ with temperature $T$ in the softmax such as $\pi_\theta(s_n|y_{<n}, x) \propto \exp(z_n/T)$ where $z_n = m_\theta(s_n|y_{<n}, x)$ is the corresponding logit. We also define $p_\theta(y|x) = \prod_{n=1}^{L} \pi_\theta(s_n|y_{<n}, x)$ the probability of sampling the output sequence $y$ from input $x$.

### 2.1 CFG DISTILLATION FOR QUALITY

**CFG.** Given a model $m_\theta$ and a partial sequence $y_{<n}$, CFG is an inference-time strategy that samples the next token by combining conditional logits $m_\theta(s_n|y_{<n}, x)$ (where $x$ is the user's prompt) and unconditional/negative logits $m_\theta(s_n|y_{<n}, x^-)$ (where $x^-$ is an empty or negative prompt):

$$z_n^{CFG_\gamma} = \gamma \cdot m_\theta(s_n|y_{<n}, x) + (1 - \gamma) \cdot m_\theta(s_n|y_{<n}, x^-). \tag{1}$$

The next-token probability distribution of the CFG-augmented policy is then $\pi_\theta^{CFG_\gamma}(s_n|y_{<n}, x) \propto \exp(z_n^{CFG_\gamma}/T)$. The guidance factor $\gamma$ controls the adherence to the prompts; higher values typically lead to higher quality (Ho & Salimans, 2022). We use $\gamma > 1$, extrapolating rather than interpolating, as done in Kreuk et al. (2023). For example, if $x$ is "Soulful jazz song." and $x^-$ is "Bad audio quality.", a larger $\gamma$ increases the degree to which the generated sequence resembles typical elements of a soulful jazz song while increasing audio quality.

**CFG distillation.** One key limitation is that CFG doubles the inference cost, since it requires the computation of two sets of logits. To eliminate this overhead, we propose an objective that directly distills the benefits of CFG into the model weights taking inspiration from similar approaches in diffusion (Luo et al., 2023; Yin et al., 2024; Saito et al., 2024; Bai et al., 2023; Novack et al., 2024). Knowledge distillation (Hinton et al., 2015) traditionally involves compressing a large teacher model into a smaller student model (Sanh et al., 2019; Agarwal et al., 2024), allowing for efficient deployment while approximating the teacher's performance. In contrast, our teacher and student share the same architecture: the teacher (with weights $\theta_{\text{init}}$ from initialisation) is simply augmented with CFG. We then follow the standard distillation approach that encourages the student to match logits from the teacher.

**On-policy distillation.** If the teacher provides the label, the remaining question is: how to sample the input data? *i.e.* the completions on which teacher's and student's logits should match. Be given a set of prompts, offline distillation uses inputs sampled from the teacher (Lin et al., 2020); in contrast, we adopt the on-policy distillation strategy from Agarwal et al. (2024), where data is online and dynamically sampled from the student itself. This reduces the train-test mismatch, *a.k.a* the exposure bias (Bengio et al., 2015), and was used in recent state-of-the-art LLMs (Gemma Team et al., 2024)[1].

**CFG distillation objective.** Overall, starting from the base pretrained initialisation $\theta_{\text{init}}$, we distill the logits obtained by $\theta_{\text{init}}$ with CFG into the weights $\theta$ without CFG by maximising:

$$\mathbb{Q}(\theta) = -\mathbb{E}_{x \sim X} \left[ \mathbb{E}_{y \sim p_\theta(\cdot|x)} \left[ \sum_{n=1}^{L} KL \left( \pi_\theta \left( \cdot | y_{<n}, x \right) || \pi_{\theta_{\text{init}}}^{CFG_\gamma} \left( \cdot | y_{<n}, x \right) \right) \right] \right]. \tag{2}$$

## 2.2 RL FOR DIVERSITY

**Reduced diversity during alignment.** Another key drawback of CFG is that it reduces the diversity of generated outputs as $\gamma$ increases. This occurs because stronger guidance encourages the model to adhere closely to the prompt, leading to outputs that are more similar to each other. This trend also emerges in our CFG distillation procedure (from Section 2.1). We confirm in Figure 2 (right) that diversity across generations from the student steadily decreases as training progresses. More generally, policy collapse is a recurring challenge for alignment strategies; for instance, Kirk et al. (2024) demonstrate that RLHF significantly reduces diversity of outputs. This limits exploration of novel and diverse contents, a crucial aspect for creative domains. To address this, we introduce a strategy encouraging the model to generate diverse samples. This requires two components, detailed below: (1) a diversity reward and (2) a diversity-enforcing algorithm.

**Diversity reward: negative cosine similarity of embeddings.** We first introduce a simple method to quantify the diversity between two generations $y_1$ and $y_2$. We embed those generations with a model $E$ and compute their cosine similarity in the embedding space. Then, the diversity is defined as $r_D(y_1, y_2) = 1 - \frac{E(y_1) \cdot E(y_2)}{\|E(y_1)\| \|E(y_2)\|}$. The key component of this diversity reward is thus the embedding model $E$. In our text-to-music application from Section 3, drawing inspiration from Futeral et al. (2024), we train a self supervised contrastive model using the semi-hard triplet loss (Schroff et al., 2015) with positive pairs coming from non-overlapping fixed-length chunks of the same audio segment. The model induces an embedding space where segments originating from the same longer audio are mapped to nearby points, while segments from different recordings are mapped further apart. Following Futeral et al. (2024), our embedding model takes as input a 4-seconds audio clip sampled at 16kHz. The model architecture is based on a compact Vision Transformer (ViT) (Dosovitskiy et al., 2021). More experimental details can be found in Appendix D.1.

**Diversity algorithm: RL for diversity.** We now leverage reinforcement learning (RL) to encourage the student policy to generate diverse samples for a given prompt. Using the diversity reward defined above, the diversity objective to maximise is:

$$\mathbb{D}(\theta) = \mathbb{E}_{x \sim X} \left[ \mathbb{E}_{y_1, y_2 \sim p_\theta(\cdot|x)} \left[ r_D(y_1, y_2) \right] \right]. \tag{3}$$

As shown in Appendix A, an unbiased estimator of $\nabla \mathbb{D}(\theta)$ is then:

$$r_D(y_1, y_2) \nabla \log p_\theta(y_1|x) \quad \text{with} \quad x \sim X, y_1, y_2 \sim p_\theta(\cdot|x). \tag{4}$$

Due to the similarity of Equation (4) with policy gradient (Sutton et al., 1999), this diversity can be optimised with standard algorithms such as REINFORCE (Williams, 1992), updating the parameters $\theta$ by increasing the likelihood of generations that lead to high (diversity) rewards. Overall, the only difference is that this gradient asks for multiple generations, matching the requirements of multi-sample RL algorithms recently used in RLHF (Ahmadian et al., 2024; Sessa et al., 2024).

## 2.3 DIVERSITY-REWARDED CFG DISTILLATION

Combining both CFG distillation from Section 2.1 and the diversity reward from Section 2.2, we maximise the following novel diversity-rewarded CFG distillation objective:

$$\mathbb{QD}(\theta) = \mathbb{Q}(\theta) + \beta \times \mathbb{D}(\theta), \tag{5}$$

---

[1]Moreover, as we also encourage the generations from the student to optimise a diversity reward, samples from the student will be readily available, making this on-policy distillation possible at no cost.

with $\beta$ the diversity hyperparameter scaling the RL diversity term to the KL quality distillation term. Low values of $\beta$ (*e.g.* 0) lead to policies focused on quality, while large values of $\beta$ (*e.g.* 15) lead to policies prioritising diversity at a slight cost in terms of quality, as visible in Figure 1 (right).

## 2.4 MODEL MERGING FOR PARETO-OPTIMAL QUALITY-DIVERSITY

Tuning the hyperparameter $\beta$ allows for some control over the quality-diversity trade-off before training, but exploring the full spectrum of possibilities would require maintaining a large set of models with different $\beta$ values. To address this, we leverage model merging, which enables combining the strengths of different models by simply interpolating their weights (Ilharco et al., 2023; Dimitriadis et al., 2023; Ramé et al., 2023). This allows us to efficiently adjust the quality-diversity trade-off at deployment time, based on the specific needs of the user or application, with only two finetunings. Specifically, we consider two models finetuned from a shared pretrained initialisation with different values of $\beta$, $\theta_q$ focused on high quality ($\beta = 0$) and $\theta_d$ focused on high diversity (high $\beta$). We then trade-off between their abilities simply by interpolating between their weights:

$$\theta_{LERP} = (1 - \lambda) \cdot \theta_q + \lambda \cdot \theta_d, \tag{6}$$

with $\lambda \in [0, 1]$. For instance, setting $\lambda$ close to 0 favors the high-quality model, generating outputs that closely adhere to the prompt, while $\lambda$ close to 1 favors the high-diversity model, resulting in more exploratory outputs. This lightweight approach uncovers a strong and steerable quality-diversity front of solutions with minimal computational cost, similar to adjusting $\gamma$ in CFG during inference.

## 3 EXPERIMENTS

The experiments below are structured to address the following research questions. (1) CFG distillation: can we distill the quality of CFG into the weights of the model, and at what cost in terms of diversity? (2) Diversity RL: can we trade-off between quality and diversity by including a diversity reward during finetuning? (3) Model merging: can we construct at inference time a strong and steerable quality-diversity front of solutions with weight interpolation?

### 3.1 TEXT-TO-MUSIC GENERATION: TASK, SETUP AND METRICS

**Task.** We explore those questions on text-to-music generation, a creative task where quality and diversity are important factors. Indeed, a single music prompt could map to many different valid generations. For instance, a gentle bossa nova piece, a relaxing acoustic guitar melody or an ethereal electronic soundscape would all be valid responses to the prompt "A peaceful sunset on a beach". This makes text-to-music generation a great testbed to study quality-diversity trade-offs.

**Model and setup.** We follow the experimental setup from Cideron et al. (2024), using their LLM transformer-based architecture, where a single autoregressive stage models the semantic and coarse acoustic features. We use their `MusicRL-R` checkpoint as the base model $\theta_{init}$, initialising all our experiments. We use the prompt dataset described in Section 4.1 from Cideron et al. (2024), combining multiple sources: synthetic prompts generated from the LaMDA model (Thoppilan et al., 2022), prompts collected via user interactions (Cideron et al., 2024), and prompts derived from MusicCaps (Agostinelli et al., 2023). For CFG, we set $\gamma = 3$ and use the negative prompt "Bad audio quality.", as it performed best on the base model (cf. Appendix B). For the diversity reward, we train a music embedding model $E$ by contrastive self-supervised learning (cf. Appendix D.1).

**Training details.** The partial generations for distillation are sampled from the student in an online fashion (Agarwal et al., 2024) with temperature $T = 0.99$. The RL algorithm is a variant of REINFORCE (Williams, 1992) with a baseline for variance reduction. We use a batch size of 128 and a learning rate of 0.00015 for all our finetunings.

**Metrics.** We evaluate the quality of the generations with two metrics: (1) the MuLan score (Agostinelli et al., 2023), a text adherence metric based on the MuLan embeddings (Huang et al., 2022), and (2) the User Preference score (Cideron et al., 2024), learned on 300k pairwise preferences of music generations to approximate the average human opinion score. We then define the general quality score as $s_{\text{quality}} = \omega \cdot s_{\text{MuLan}} + s_{\text{User Preference}}$, using $\omega = 5$ to enforce a similar range of variations for the two scores. Critically, those metrics are not used during training. In Section 3.5, we confirm the insights obtained from this quality metric through human evaluation.

## 3.2 CFG DISTILLATION FOR QUALITY

We want to validate whether or not it is possible to match by distillation (at training time) the performances of CFG-augmentation (at inference time) without its inference cost overhead.

**CFG distillation optimisation.** Figure 2 (left) shows the evolution of the KL divergence from Equation (2) between the policy and the CFG-augmented policy along training. When the objective is only to minimise the KL (*i.e.* $\beta = 0$), the KL decreases to its lowest value, implying that the CFG-free model learns to approximate the CFG logits. $\beta$ is scheduled to linearly increase until step 1000, which explains why the KL increases after this step for $\beta > 0$.

**CFG distillation improves quality.** Figure 2 (middle) shows that training significantly improves quality matching the performances of the base model augmented with CFG and $\gamma = 3$. This suggests that CFG's quality can be effectively distilled into the model's weights, a finding confirmed by human evaluations in Section 3.5.

**CFG distillation reduces diversity.** Figure 2 (right) shows that the CFG-distilled model ($\beta = 0$) suffers from a decrease in diversity compared to its initialisation, the CFG-free base model. Notably, after only 1k training steps, CFG-distillation reaches the same level of diversity (around 0.37) than the CFG-augmented base model (grey horizontal line for $\gamma = 3$). As a side note, Figure 1 (right), Figure 4 (right) and Figure 6 (in Appendix B) show that higher values of $\gamma$ for CFG further reduce diversity. In the next subsection, we assess the efficiency of our diversity reward to promote diversity.

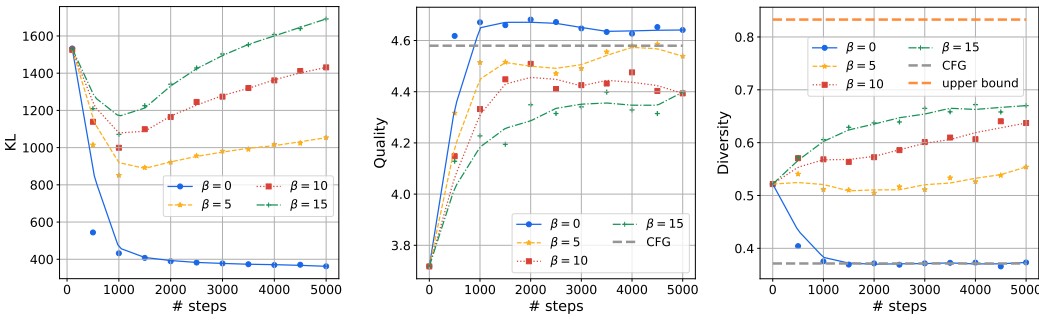

Figure 2: **Left.** Evolution of the KL divergence between the CFG-distilled student and the CFG-augmented teacher along training. GKD distillation alone ($\beta = 0$) decreases the KL between the two policies. **Middle.** Evolution of the quality along training, showing improved quality for all selected values of $\beta$. **Right.** Evolution of the diversity across generations along training, showing that CFG distillation alone reduces diversity, but that using a diversity reward ($\beta \neq 0$) can actually increase it. The "CFG" line shows the quality/diversity performance of the CFG-augmented base model serving as a teacher. The "upper-bound" line indicates the mean diversity of two generations (from the base model) for two different prompts.

## 3.3 RL FOR DIVERSITY

**Rewarding diversity.** We now analyse the results obtained when including the diversity reward from Equation (3), scaled by the hyperparameter $\beta \in \{0, 5, 10, 15\}$ in the joint objective. As visible in Figure 2 (right), this increases the diversity across generations along training. For $\beta = 10$ or $\beta = 15$, diversity actually gets larger, and closer to the mean diversity between two generations for two different prompts, denoted as an empirical "upper-bound".

**Quality-diversity trade-off.** Figure 2 (left) shows that these diversity gains come at the cost of increased KL. This is because higher values of $\beta$ put more emphasis on the diversity term from Equation (3), hindering the minimisation of the KL distillation loss from Equation (2). As a consequence, policies finetuned with larger values of $\beta$ have lower quality in Figure 2 (middle). Those insights are summarised in Figure 1 (right), where the quality-diversity trade-off is plotted along training for the different values of $\beta$.

## 3.4 MODEL MERGING FOR PARETO-OPTIMAL QUALITY-DIVERSITY

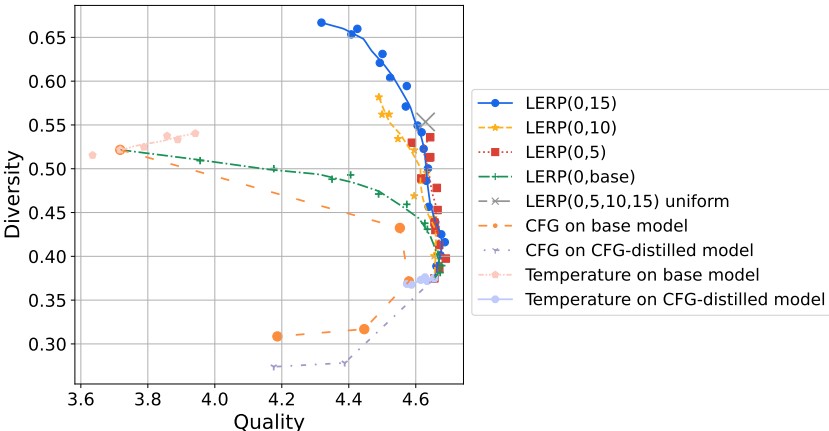

Figure 3: Quality-diversity trade-off for multiple strategies. The first four lines linear interpolate (LERP) between the quality-focused model ($\beta = 0$) and more diverse models (those trained with $\beta > 0$, or the base model), sliding $\lambda$ between 0 and 1 with a step of 0.05. We also report the performance from the uniform ($\lambda = \frac{1}{4}$) averaging of the four models finetuned with different $\beta$, denoted as "LERP$(0, 5, 10, 15)$ uniform". We include inference-time baseline strategies — CFG (when varying $\gamma$) and temperature sampling (when varying the temperature $T$) — applied either on the base model or on the CFG-distilled model.

**Quality-diversity trade-off.** In Figure 3 we interpolate between the model with highest quality and the model with highest diversity: by sliding the coefficient $\lambda$ from 0 (only the quality model) to 1 (only the diversity model), we construct fronts of solutions that surpass the performance of individual models finetuned for intermediate $\beta$ values. In particular, interpolating towards the solution with maximum diversity ($\beta = 15$) yields a strong and steerable Pareto front describing the full range of possible trade-offs. This indicates that model merging achieves a higher quality for a given level of diversity, and vice versa, consistent with previous studies that show how merged models perform better than their non-merged counterparts (Wortsman et al., 2022a; Ramé et al., 2022). This is achieved with minimal overhead: only two RL finetuning runs are required, and merging the weights involves negligible computational cost. If only one finetuning is possible, then linearly interpolating towards the base initialisation can also help, by recovering features from pretraining (Wortsman et al., 2022b); in particular we note that "LERP$(0, \text{base})$" outperforms the CFG-augmented base model. If more compute is available for training and we can do four finetunings, then merging them uniformly performs even better, as visible in Figure 3 where the grey cross for "LERP$(0, 5, 10, 15)$ uniform" is (slightly) above the other fronts. This suggests that we can merge an arbitrary number of models.

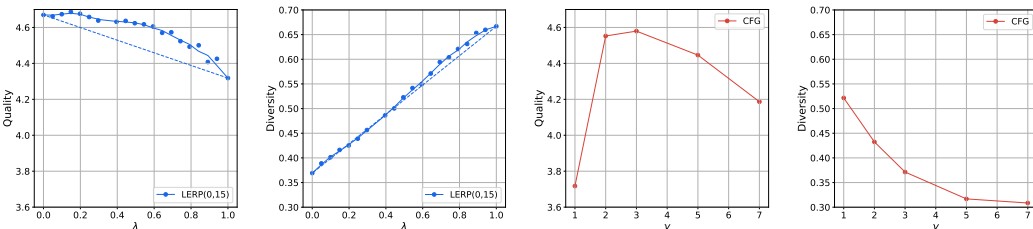

Figure 4: **Left.** Linear interpolation between the weights of a model focused on quality ($\beta = 0$) and a model focused on diversity ($\beta = 15$), sliding the interpolating coefficient $\lambda$ between 0 and 1. The dashed diagonal represents the expected values if abilities were traded-off linearly between those two models. While the diversity stays close to the diagonal, the quality remains above it, showing the benefits of model merging. **Right.** For comparison, we also include the results for CFG when sliding $\gamma$ between 1 and 7, performing worse than merged models.

**Baselines.** Figure 3 also displays the results for two inference-time baselines: CFG and temperature sampling. Applied to the base model, these strategies are Pareto-dominated by CFG-distilled models with diversity rewards. When applied to the already CFG-distilled model, they do not significantly improve quality nor diversity.

**Quality-diversity as a function of $\lambda$.** Figure 4 (left) clarifies the effect of weight interpolation on quality and diversity as we slide $\lambda$ between 0 and 1. Diversity stays close to the diagonal (representing the expected diversity) while the quality is consistently above the diagonal. This highlights the ability to significantly increase diversity with only minor quality compromises via model merging.

## 3.5 HUMAN EVALUATION

**Protocol.** To conclude our experiments, we validate via human evaluation the improved quality-diversity trade-offs. We evaluate five models: the base model (we expect low quality but high diversity), the base model with CFG $\gamma = 3$ (high quality but low diversity), the CFG-distilled model for $\beta = 0$ (high quality but low diversity) and $\beta = 15$ (medium quality and high diversity), and finally LERP(0, 15) merging uniformly with $\lambda = 0.5$ the two previous models (high quality and medium diversity). For quality (*i.e.* acoustic quality, text adherence, and musicality), we use the same evaluation protocol as in Cideron et al. (2024): the raters see two generations from different models coming from the same text prompt and rate them on a scale from 1 to 5. We use 100 different prompts and each one is seen by 3 different raters. For diversity, we rely on a similar protocol except that the raters see two pairs of music clips generated from the same text prompts and are rate how diverse the pairs of generations are. We use 50 different prompts and each one is rated by 3 different raters.

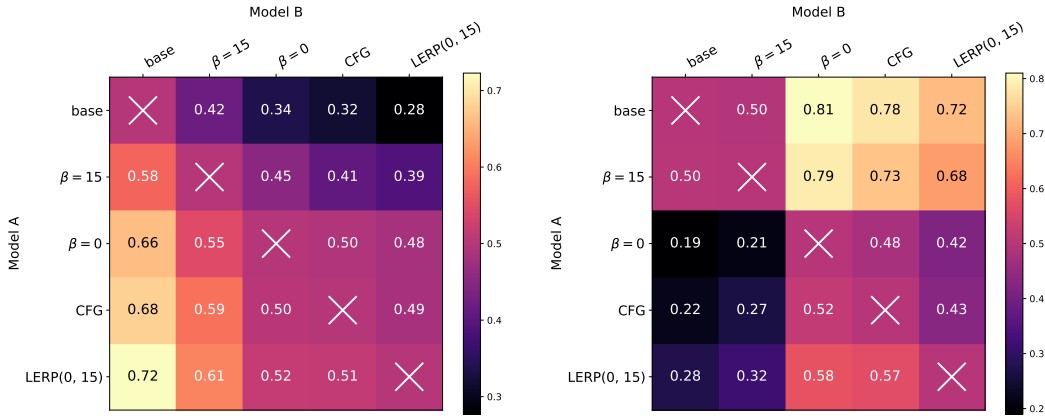

Figure 5: **Left.** Side-by-side human evaluation for quality. **Right.** Side-by-side human evaluation for diversity. The score corresponds to the win rate of model A over model B, computed as $\frac{W+T/2}{W+L+T}$ with $W$ the number of wins of A over B, $T$ the number of ties, $L$ the number of losses of A against B. This confirms that our approach improves the quality-diversity trade-off. For instance, the merged model LERP(0, 15) generates music with higher diversity than the CFG-augmented base model ($\gamma = 3$) in 57% of the comparisons, while being rated as more qualitative half of the time (51%).

**Human evaluation for quality.** Figure 5 (left) presents the side-by-side win rate for quality. The CFG-distilled model (*i.e.* $\beta = 0$) performs on par with the CFG-augmented base model (*i.e.* CFG) with a win rate of 0.50; they both outperform the base model without CFG (first line), with 0.66 and 0.68 win rate respectively. LERP(0, 15) achieves win rate above 0.50 against all models.

**Human evaluation for diversity.** Figure 5 (right) presents the side-by-side win rate for diversity. The CFG-augmented base model (*i.e.* CFG) and the CFG-distilled model (*i.e.* $\beta = 0$) are also evaluated similarly in terms of diversity (0.52 win rate for the former); yet, they are consistently seen less diverse than the three other models. Notably, the CFG-distilled model with diversity reward (*i.e.* $\beta = 15$) has win rate of 0.73 and 0.79 against them. As a side note, this $\beta = 15$ model performs on par with the base model (0.50 win rate), though the quantitative metrics from Figure 2 (right) suggested that it was actually more diverse; in Section 5, we relate this discrepancy to the standard reward hacking (Amodei et al., 2016; Gao et al., 2023) phenomenon of RL.

**Human evaluation for quality-diversity trade-off.** Human evaluations confirm the effectiveness of our approach in improving the quality-diversity trade-off. The diversity-focused model ($\beta = 15$) exhibits higher quality than the base model while maintaining comparable diversity. Importantly, the merged model LERP(0, 15) exhibits higher diversity than the CFG-augmented base model while maintaining comparable quality, and without incurring the additional inference cost of CFG.

**Qualitative analysis.** To facilitate qualitative analysis, we host music generated from all evaluated models on the website google-research.github.io/seanet/musiclm/diverse_music. An examination of generic prompts like "Rock song." confirms that CFG improves quality (*i.e.* less acoustic artifacts, better musicality) but reduces diversity, as generated guitars tend to be very similar across different generations. In contrast to the CFG-augmented model, the $\beta = 15$ model generates a wider variety of rhythms while still demonstrating a clear improvement in quality over the base model. On prompts like "Opera singer" or "Male choir harmony", the quality-focused model $\beta = 0$ generates conventional outputs, while the diverse model $\beta = 15$ produces more unconventional and creative results, sometimes leading to unexpected elements like unusual instrumentation (*e.g.* drums). By averaging the weights of these two models (LERP), we can effectively balance these qualities, generating high-quality music that is both faithful to the prompt and more creative.

## 4 RELATED WORK

**Classifier-free guidance (CFG).** Initially introduced for diffusion models (Ho & Salimans, 2022) and later adapted for autoregressive LLMs (Gafni et al., 2022; Sanchez et al., 2023; Wings, 2022), CFG has found widespread application in various generative domains, including image (Yu et al., 2022; Saharia et al., 2022; Rombach et al., 2022; Nichol et al., 2021; Ramesh et al., 2022), video (Blattmann et al., 2023; Ho et al., 2022), and audio (Kreuk et al., 2023; Copet et al., 2023) generation. However, the detrimental impact of CFG on diversity is well-documented (Dhariwal & Nichol, 2021; Kreuk et al., 2023; Nichol et al., 2021), limiting its application when exploration is key.

**Distillation.** Knowledge distillation (Hinton et al., 2015) is emerging as a powerful technique to train state-of-the-art models (Gemma Team et al., 2024). By transferring knowledge from a teacher, the student can perform better than with standard training on the same data (Sanh et al., 2019; Lin et al., 2020; Gu et al., 2023). In the context of diffusion models, CFG distillation was applied to drastically reduce the inference time (Meng et al., 2023; Luo et al., 2023; Yin et al., 2024; Saito et al., 2024; Bai et al., 2023; Novack et al., 2024). In our work, we apply the CFG distillation idea to LLMs where we employ a single-stage on-policy distillation procedure (Agarwal et al., 2024) to distill a CFG-augmented LLM, while introducing a novel diversity-promoting RL algorithm and model merging for improved quality-diversity trade-off.

**Quality-diversity in LLMs.** Zhang et al. (2021) compare the quality-diversity trade-offs of various inference-time strategies for LLMs, including temperature sampling, top-k sampling (Fan et al., 2018), and nucleus sampling (Holtzman et al., 2020). These methods perform similarly except when quality is prioritized over diversity, where nucleus sampling performs best. Regarding finetuning strategies, Kirk et al. (2024); Mohammadi (2024); Chaudhari et al. (2024); Li et al. (2024) have shown that RLHF can negatively impact the diversity of generated text, often measured with metrics like BLEU. To the best of our knowledge, we are the first to introduce an RL algorithm to optimize diversity, which could potentially also solve those reductions in diversity in RLHF. In contrast, existing quality-diversity algorithms (Lehman & Stanley, 2011; Mouret & Clune, 2015; Cully et al., 2015; Cideron et al., 2020; Ding et al., 2024) aim at finding a population of agents with both high-quality and diverse behaviors. Most similarly, Li et al. (2016); Zhang et al. (2018) also tried to increase the diversity across generations produced by a single agent, but measured by the number of distinct $n$-grams, and optimised with objectives based on mutual information (Shannon, 1948).

**Model merging for Pareto-optimality.** Model merging via weight averaging (Utans, 1996) has two main applications in deep learning. First, it increases generalisation by reducing variance (Wortsman et al., 2022a; Ramé et al., 2022) and memorisation (Lin et al., 2024; Ramé et al., 2024). Second, it combines their strengths (Ilharco et al., 2023; Dimitriadis et al., 2023; Wang et al., 2024), as employed in Ramé et al. (2023) for multi-objective RL, where policies finetuned with different rewards are interpolated. Similarly, we interpolate between policies where only one of them is rewarded for diversity. Despite recent theoretical efforts to explain the empirical success of model merging (Ferbach et al., 2024; Ramé et al., 2024), a complete understanding remains elusive.

**Music generation.** Diffusion models (Huang et al., 2023; Schneider et al., 2023; Liu et al., 2023) and Transformers (Agostinelli et al., 2023; Copet et al., 2023) are now the state-of-the-art architectures for music generation. In this work, we leverage the Transformer-based approach from Agostinelli et al. (2023), casting music generation as a categorical prediction task in the discrete token space provided by a neural audio codec (Zeghidour et al., 2022; Défossez et al., 2022). RL-finetuning for music generation has previously been explored in Jaques et al. (2017); Guimaraes et al. (2017); Kotecha (2018); Latif et al. (2023); Cideron et al. (2024). In contrast, we are the first to distill CFG, to enforce diversity through RL, to apply model merging or to improve the quality-diversity trade-off for music.

## 5 DISCUSSIONS AND LIMITATIONS

**Amplification and distillation.** In this work, we leverage a teacher-student framework where the teacher is an augmented version of the student model itself. This allows to distill the quality improvements obtained from a potent but expensive inference strategy, eliminating the overhead at deployment. This echoes the principles of iterated distillation and amplification (IDA) (Cotra, 2018), where the model iteratively learns on data generated by an augmented version of itself. Examples of this can be seen in AlphaGo (Silver et al., 2016), achieving superhuman performance by distilling the knowledge of MCTS (Kocsis & Szepesvári, 2006), or in recent LLM works (Wang et al., 2023; Yu et al., 2024), distilling "System 2" into "System 1" by imitating offline predictions obtained from Chain-of-Thought (Wei et al., 2022). Given the effectiveness of scaling inference-time strategies (Snell et al., 2024), we anticipate wider adoption of such amplification-then-distillation techniques.

**Extension to other models or modalities.** The three main components of our approach — distillation of an inference strategy, RL with a diversity reward, and model merging — could be readily adapted to other generative architectures. For instance, model merging is already a popular strategy for diffusion models (Purplekeyboard, 2022; Biggs et al., 2024). Additionally, while our focus is on text-to-music, similar strategies could be applied to other setups involving text, image or video generation. For applications where retaining the CFG coefficient is crucial our approach can easily be extended with methods like CLP (Wang et al., 2024) . It would only require to swap the underlying RL algorithm with the CLP algorithm.

**Diversity measures.** We used negative cosine similarity between embeddings as our diversity measure. Critically, we demonstrate in Appendix D.2 its superior correlation with human perception of diversity compared to token-level entropy: we suspect this is because diversity should be assessed at the sentence level for creative tasks. Yet, qualitative inspection suggests that this diversity measure can still be hacked (Amodei et al., 2016; Gao et al., 2023). For example, models finetuned to maximise diversity sometimes generate excessive background drums, likely due to a bias of the underlying embedding model $E$. This could be changed by making $E$ more invariant to background drums; alternatively, inspired by Ding et al. (2024), we could learn directly a human feedback (or AI-feedback) diversity embedding, by asking humans (or LLMs) to rate the similarity of generations. More broadly, relying on a single diversity metric may be insufficient, as diversity can manifest in various ways (Levy et al., 2024); in music, diversity could focus on variations in voice, key, tempo, or other elements; in NLP, a summarization task may prioritise syntactic diversity over semantic diversity. By incorporating multiple diversity rewards, each targeting a specific variation, and then leveraging model merging, we could achieve fine-grained control over diversity at deployment time.

## 6 CONCLUSION

In this work, we introduced diversity-rewarded CFG distillation, a novel finetuning approach to enhance the quality-diversity trade-off in generative models. First, we online distilled CFG, eliminating its computational overhead at inference time. Second, to preserve and even enhance diversity across generations, we incorporated an RL procedure that optimises a diversity reward based on similarity embeddings. Third, we leveraged model merging to enable dynamic control over the quality-diversity trade-off at deployment time. Through extensive experiments on text-to-music generation, we demonstrated the validity of our strategy, with our finetuned-then-merged model performing best according to human evaluations. We believe that our work provides a promising foundation for future research exploring tasks where alignment and creativity are key, and that it can be easily extended to setups beyond text-to-music generation.

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

# Diversity-Rewarded CFG Distillation

## Supplementary material

## A  POLICY GRADIENT FOR THE DIVERSITY REWARD

In this appendix we consider the general case of optimising diversity across $N$ generations, where:

$$r_D(y_1, \ldots, y_N) = \frac{\sum_{i,j \in \{1,\ldots,N\}, i \neq j} r_D(y_i, y_j)}{\sum_{i,j \in \{1,\ldots,N\}, i \neq j} 1}. \tag{7}$$

The key originality is that this reward considers multiple trajectories simultaneously, while the traditional policy gradient from Sutton (2018) considers a single trajectory. We make the theoretical connection below.

Let's consider a symmetric multi-trajectories reward, where the expected reward is defined as follows:

$$J(\theta) = \frac{1}{N} \mathbb{E}_{x \sim X} \left[ \mathbb{E}_{y_1,\ldots,y_N \sim p_\theta(\cdot|x)} \left[ r_D(y_1, \ldots, y_N) \right] \right]. \tag{8}$$

If we expand the inner expectation:

$$\mathbb{E}_{y_1,\ldots,y_N \sim p_\theta(\cdot|x)} \left[ r_D(y_1, \ldots, y_N) \right] = \sum_{y_1,\ldots,y_N} r_D(y_1, \ldots, y_N) \prod_{i=1}^{N} p_\theta(y_i|x). \tag{9}$$

Then we take the gradient:

$$\nabla \mathbb{E}_{y_1,\ldots,y_N \sim p_\theta(\cdot|x)} \left[ r_D(y_1, \ldots, y_N) \right] = \nabla_\theta \left[ \sum_{y_1,\ldots,y_N} r_D(y_1, \ldots, y_N) \prod_{i=1}^{N} p_\theta(y_i|x) \right] \tag{10}$$

$$= \sum_{y_1,\ldots,y_N} \sum_{j=1}^{N} \nabla p_\theta(y_j|x) r_D(y_1, \ldots, y_N) \prod_{i=1,i \neq j}^{N} p_\theta(y_i|x) \tag{11}$$

$$= \sum_{y_1,\ldots,y_N} \prod_{i=1}^{N} p_\theta(y_i|x) \left[ \sum_{j=1}^{N} \nabla \log p_\theta(y_j|x) \right] r_D(y_1, \ldots, y_N) \tag{12}$$

$$= \sum_{j=1}^{N} \sum_{y_1,\ldots,y_N} \nabla \log p_\theta(y_j|x) r_D(y_1, \ldots, y_N) \prod_{i=1}^{N} p_\theta(y_i|x) \tag{13}$$

$$= N \sum_{y_1,\ldots,y_N} \nabla \log p_\theta(y_1|x) r_D(y_1, \ldots, y_N) \prod_{i=1}^{N} p_\theta(y_i|x). \tag{14}$$

For Equation (12), we used $\nabla \log x = \nabla x / x$. For Equation (14), we used the symmetry of $r_D$ and change of variable. Hence, a non biased estimator of the policy gradient for $\nabla J(\theta)$ is:

$$r_D(y_1, \ldots, y_N) \nabla \log p_\theta(y_1|x) \text{ where } x \sim X, y_1, \ldots, y_N \sim p_\theta(\cdot|x). \tag{15}$$

Due to the similarity of Equation (15) with the estimator of policy gradient for single-trajectory rewards, the diversity reward can be optimised with the usual policy gradient algorithms. As an example, the sampled gradient with a batch size $B$ of the vanilla REINFORCE algorithm (Williams, 1992) would become $\frac{1}{BN} \sum_{i=1}^{B} \sum_{j=1}^{N} r_D(y_{i,1}, \ldots, y_{i,N}) \nabla \log p_\theta(y_{i,j}|x_i)$ with $x_i \sim X, y_{i,j} \sim p_\theta(\cdot|x_i)$.

## B  CFG FOR TEXT-TO-MUSIC GENERATION

Figure 6 shows the impact of CFG when applied at inference time on the `MusicRL-R` base model. Specifically, we plot the quality-diversity front of solutions when sliding the adherence hyperparameter $\gamma$ for unconditional and negative prompting. We observe that CFG significantly decreases diversity going from 0.5 (for $\gamma = 1$) to 0.3 (for $\gamma = 7$). Critically, we also see that the front of solutions for negative prompting with "Bad audio quality." is above the one revealed by unconditional prompting. This specific negative prompt, inspired by those used in image generation (Mostaque, 2023), was selected based on preliminary experiments where it outperformed other negative prompt candidates. Finally, for quality the best $\gamma$ is 3, which we use throughout this work.

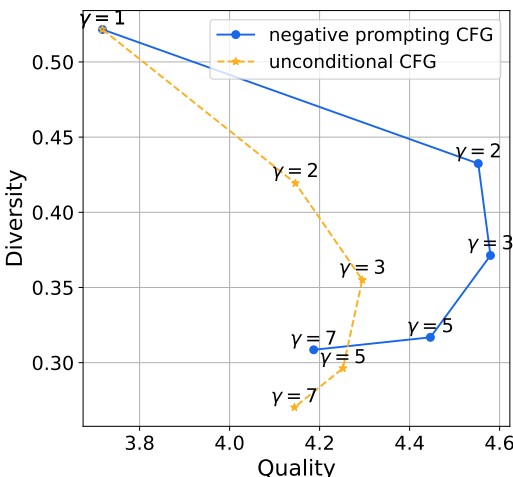

Figure 6: Effect of CFG on quality and diversity as a function of $\gamma$. The quality is greatly improved at the expense of diversity. Negative prompting outperforms unconditional prompting.

Figure 7 shows the effect of CFG on the User Preference score and the MuLan score. Unconditional CFG improves the User Preference score and the MuLan score which indicates that the generations are both of better quality and of better adherence to the text. For negative prompting CFG, the User Preference score gain is doubled compared to unconditional. However, the MuLan score decreases which can be explained as the generations are biased towards good acoustic quality. With negative prompting CFG, the model tends to always produce high quality audio even when the prompt does not explicitly mention it.

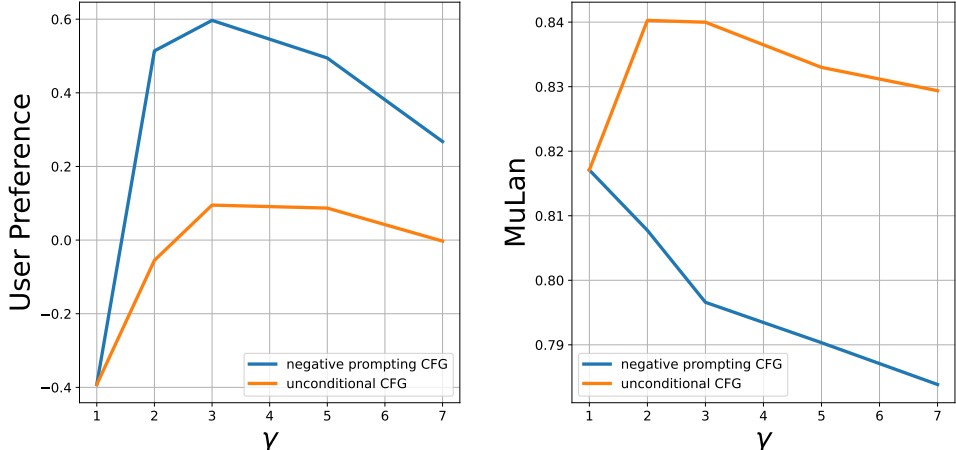

Figure 7: Effect of CFG on the User Preference score and the MuLan score. Negative prompting CFG improves twice as much the User Preference score. For unconditional CFG, the MuLan score which reflects the text adherence increases which is an expect effect of CFG. For negative prompting CFG, there is a drop of MuLan which can be explained as we biased the generations towards 'good audio quality' which is something that might not be present in the text prompt.

## C  DETAILS ON THE EVALUATION METRICS

Figure 8 shows the User Preference and the MuLan score for the different models. Figure 8 shows that (1) $\beta = 0$ matches the performances of CFG on both metrics and (2) increasing the $\beta$ coefficient primarily impacts the text adherence score while keeping the User Preference score nearly constant. It shows that that our approach improves the diversity by reducing the text adherence without impacting the audio quality.

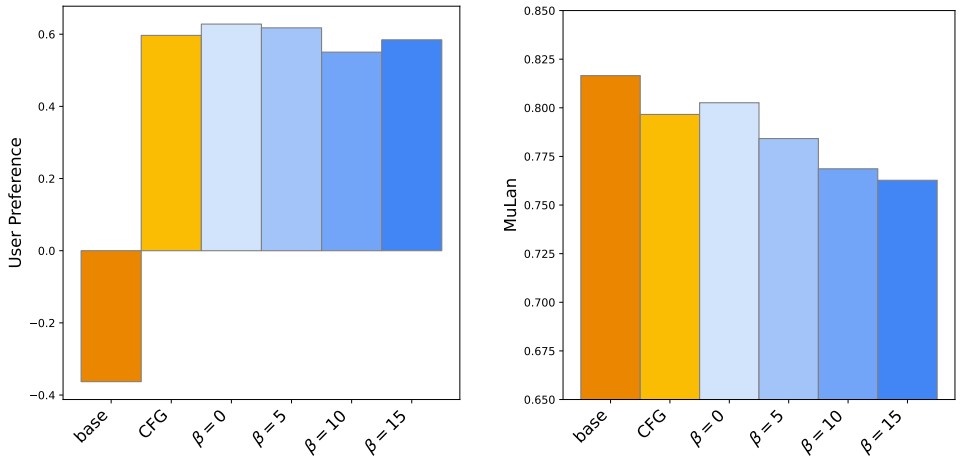

Figure 8: Comparison of the performances of the models on the User Preference score and the MuLan score. $\beta = 0$ matches the performances of CFG. Increasing the $\beta$ coefficient primarily impacts the text adherence score while keeping the User Preference score nearly constant. It shows that our approach improves the diversity by reducing the text adherence without impacting the audio quality.

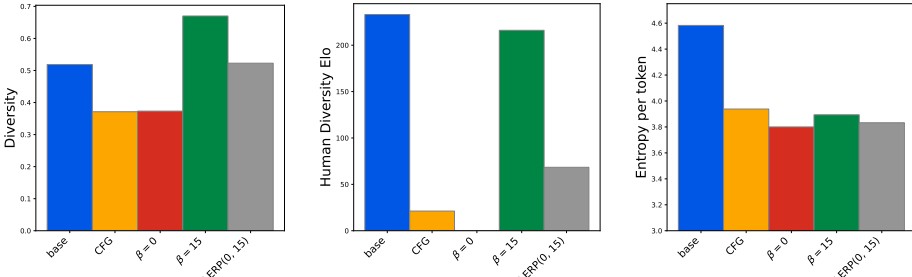

Figure 9: Comparison between different measures of diversity; the one provided by our embedding model (left), the Elo rankings provided by human evaluation of diversity (middle) and the entropy per token (right). CFG at inference time (yellow) reduces all those measures of diversity. Models trained with our RL diversity reward ($\beta = 15$ in green) have increased diversity according to our embedding model and human evaluation, but not in terms of entropy.

## D  DIVERSITY IN MUSIC

### D.1  EMBEDDING MODEL FOR DIVERSITY

**Diversity in music.** The objective of our diversity reward is to estimate the dissimilarity between two music segments. Diversity, especially in music, is a multifaceted concept encompassing various aspects such as instrumentals, melodies, tempo. In this work we conceptualise diversity through the following question: *Are these two audio excerpts derived from the same audio recording or not?*

**Embedding.** Drawing inspiration from Futeral et al. (2024), we train a self supervised contrastive model with positive pairs coming from non-overlapping fixed-length chunks of the same audio segment. The model induces an embedding space where segments originating from the same longer audio are mapped to nearby points, while segments from different recordings are mapped further apart. This is useful because cosine similarity in the embedding space can quantify the similarity between two audio segments.

**Details.** Following Futeral et al. (2024), our embedding model takes as input a 4-seconds audio clip sampled at 16kHz. We generate the mel-spectrogram using a window size of 512 samples, a hop length of 256 samples, and 156 frequency bins. This results in 376 time frames, each with 156 dimensions. The model architecture is founded on a compact Vision Transformer (ViT) (Dosovitskiy et al., 2021), comprising 12 layers. Each layer incorporates 6 attention heads, an embedding dimension of 512, and a feed-forward layer with a dimension of 1024 (totaling 25 million parameters). The final embeddings are averaged over time and then projected into a 192-dimensional space. For training, we employ the semi-hard triplet loss (Schroff et al., 2015) with a total batch size of 3840, for 200,000 steps, using 16 kHz audio excerpts sourced from the same training dataset as Agostinelli et al. (2023). Additionally, we augment the training data with noise, tempo and pitch shifts.

### D.2  COMPARISON OF OUR DIVERSITY METRIC WITH HUMAN EVALUATION AND ENTROPY

We compare the correlation between our diversity metric, the Elo score computed from human diversity evaluation in Section 3.5, and the entropy per generated token (which is a commonly used proxy for diversity (Zhang et al., 2021)). Figure 9 highlights that our diversity measure has higher correlation with the human raters' notion of diversity than entropy. As an example, $\beta = 15$ scores high both in terms of Elo score and our diversity metric, while the entropy per token is as low as CFG (which is a low diversity approach). Yet, our measure of diversity remains imperfect and thus can be hacked, as further discussed in Section 5.

### D.3 OPTIMISATION OF THE DIVERSITY REWARD ACCORDING TO THE NUMBER OF GENERATIONS.

To compute the diversity reward $r_D(y_1, \ldots, y_N)$, one important parameter is $N$ the number of generations. Figure 10 shows the quality (left) and diversity (right) obtained when varying the number of generations $N \in \{2, 4, 8\}$ while keeping constant the batch size at 128. Figure 10 shows that $N = 2$ is the best hyperparameter to optimise the diversity. Hence, we used $N = 2$ throughout our experiments.

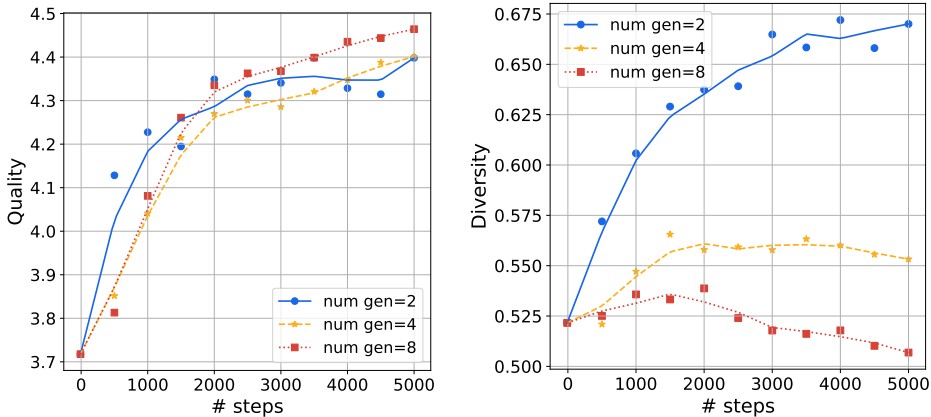

Figure 10: Comparison between different values of $N$ the number of generations per prompt. In this experiment we ran the same experimental setup ($\beta = 15$, batch size$= 128$) with $N \in 2, 4, 8$. The results show that while the quality score is roughly the same for every values of $N$ (left), the diversity score is best optimised with $N = 2$ (right).

### D.4 VENDI SCORE VS MEAN SIMILARITY

For the 101 prompts used for the quality human evaluation, we generated 8 music clips and we computed the diversity of these music clips with two measures (1) a simple 1-mean computed on the similarity matrix, and (2) the Vendi score (Friedman & Dieng, 2022) computed on the similarity matrix. Table 1 shows that both measures achieve the same ranking in term of diversity.

Table 1: Comparison of the diversity score computed on 8 generations generated with the prompts from the human evaluation. The diversity score is either computed with a simple 1-mean of the similarity matrix or with the Vendi Score (Friedman & Dieng, 2022) computed on the similarity matrix.

| Model | Vendi Score | 1 - Mean |
|---|---|---|
| $\beta = 15$ | 2.91 | 0.60 |
| base | 2.40 | 0.46 |
| LERP(0, 15) | 2.35 | 0.44 |
| $\beta = 0$ | 1.95 | 0.32 |
| CFG | 1.93 | 0.32 |

## E    DETAILS FOR HUMAN EVALUATION

Table 2 and Table 3 show the details results of the side by side human evaluation for respectively quality and diversity. The p-values for two statistical tests, a paired t-test and the Wilcoxon signed rank test (Rey & Neuhäuser, 2011) are displayed in these tables. Our results are statistically significant according to both statistical tests.

Table 2: Details of the side-by-side quality human evaluation. Two statistical tests were performed on the results: the paired t-test and the Wilcoxon signed-rank test (Rey & Neuhäuser, 2011). The p-values for these two tests are displayed in the last two columns.

| Model A | Model B | #WINSA | #WINSB | #DRAWS | paired t-test | Wilcoxon test |
|---|---|---|---|---|---|---|
| base | CFG | 59 | 168 | 76 | 6.5e-14 | 1.0e-12 |
| base | $\beta = 0$ | 62 | 161 | 80 | 6.3e-13 | 5.0e-12 |
| base | $\beta = 15$ | 82 | 130 | 91 | 2.8e-04 | 3.0e-04 |
| base | LERP(0, 15) | 49 | 184 | 70 | 1.7e-22 | 1.4e-19 |
| CFG | $\beta = 0$ | 107 | 104 | 92 | 7.4e-01 | 7.3e-01 |
| CFG | $\beta = 15$ | 133 | 78 | 92 | 2.7e-05 | 3.1e-05 |
| CFG | LERP(0, 15) | 96 | 105 | 102 | 8.0e-01 | 8.2e-01 |
| $\beta = 0$ | $\beta = 15$ | 123 | 95 | 85 | 1.9e-02 | 1.8e-02 |
| $\beta = 0$ | LERP(0, 15) | 99 | 109 | 95 | 5.1e-01 | 5.1e-01 |
| $\beta = 15$ | LERP(0, 15) | 74 | 140 | 89 | 9.0e-06 | 1.4e-05 |

Table 3: Details of the side-by-side diversity human evaluation. Two statistical tests were performed on the results: a paired t-test and the Wilcoxon signed-rank test (Rey & Neuhäuser, 2011). The p-values for these two tests are displayed in the last two columns.

| Model A | Model B | #WINS0 | #WINS1 | #DRAWS | paired t-test | Wilcoxon test |
|---|---|---|---|---|---|---|
| base | CFG | 107 | 23 | 20 | 9.6e-14 | 1.8e-11 |
| base | $\beta = 0$ | 114 | 21 | 15 | 9.4e-20 | 1.0e-15 |
| base | $\beta = 15$ | 56 | 57 | 37 | 6.2e-01 | 6.4e-01 |
| base | LERP(0, 15) | 95 | 28 | 27 | 1.5e-08 | 2.4e-07 |
| CFG | $\beta = 0$ | 62 | 56 | 32 | 5.8e-01 | 5.3e-01 |
| CFG | $\beta = 15$ | 30 | 100 | 20 | 2.1e-11 | 3.6e-10 |
| CFG | LERP(0, 15) | 53 | 74 | 23 | 2.3e-02 | 1.6e-02 |
| $\beta = 0$ | $\beta = 15$ | 22 | 108 | 20 | 2.6e-16 | 1.0e-13 |
| $\beta = 0$ | LERP(0, 15) | 51 | 75 | 24 | 3.9e-02 | 5.3e-02 |
| $\beta = 15$ | LERP(0, 15) | 92 | 37 | 21 | 2.6e-07 | 4.9e-07 |

## F  CLAP SCORE METRICS

For the 101 prompts used for the quality human evaluation, we generated music clips and computed the CLAP score (Elizalde et al., 2023). As we generated 8 music clips per prompts, we computed a mean CLAP score and a standard deviation. Table 4 shows the CLAP score for the different models. All models perform equally.

Table 4: Clap scores computed on the 101 prompts used for the quality human evaluation. All models perform equally.

| Model | Clap Score |
|---|---|
| Beta=0 | $0.52 \pm 0.11$ |
| Beta=15 | $0.52 \pm 0.11$ |
| LERP(0, 15) | $0.52 \pm 0.11$ |
| Base | $0.53 \pm 0.11$ |
| CFG | $0.53 \pm 0.11$ |

## G  FAD SCORE

Table 5 shows the FAD score computed on MusicCaps (Agostinelli et al., 2023) with two audio embedding backbones namely CLAP (Elizalde et al., 2023) and VGGish (Hershey et al., 2017) and

the FAD score computed on SongDescriber (Manco et al., 2023). We see a similar pattern for all the variations. Without diversity (*i.e.* $\beta = 0$) CFG Distillation matches the performance of CFG. Then, as we increase diversity, FAD decreases consistently. However, there is an inconsistency as the base model has a lower FAD than CFG which is contradicted with the quality human evaluation that shows that CFG is better than base. The inconsistency on the FAD score were reported in previous work (Copet et al., 2023; Gui et al., 2024).

Table 5: FAD on MusicCaps (Agostinelli et al., 2023) computed with either CLAP (Elizalde et al., 2023) or VGGish (Hershey et al., 2017) and FAD with CLAP on SongDescriber (Manco et al., 2023). We see a similar pattern for all the variations. Without diversity (*i.e.* $\beta = 0$) CFG Distillation matches the performance of CFG. Then, as we increase diversity, FAD decreases consistently. However, there is an inconsistency as the base model has a lower FAD than CFG which is contradicted with the quality human evaluation that shows that CFG is better than base.

| Dataset | MusicCaps | MusicCaps | SongDescriber |
|---|---|---|---|
| Audio Embedding | FAD w/ VGGish | FAD w/ CLAP | FAD w/ CLAP |
| base | 3.759 | 0.15 | 0.11 |
| CFG | 4.371 | 0.21 | 0.13 |
| $\beta = 0$ | 4.357 | 0.21 | 0.13 |
| LERP(0, 15) | 4.986 | 0.23 | 0.13 |
| $\beta = 15$ | 6.437 | 0.26 | 0.15 |

