# OpenReview forum: "Diversity-Rewarded CFG Distillation"
_ICLR.cc/2025/Conference — ICLR 2025 Poster_

### Official Review · Reviewer_suju · 2024-10-26

**Soundness:** 2
**Presentation:** 4
**Contribution:** 3
**Rating:** 6
**Confidence:** 3

**Summary:**

The paper proposes a distillation based method to improve text to music models (specifically MusicLM and MusicRL). The student network learns the CFG-modified logits of the teacher model, and therefore running the teacher model with CFG is approximated with running the student network directly, which reduces inference compute to half. To address less diversity caused by CFG and by the CFG distillation, the paper proposes to maximize the diversity computed from inter-similarities during CFG distillation. The paper also finds directly interpolating two model weights computed from different $\beta$ leads to better quality-diversity tradeoff.

**Strengths:**

The paper is extremely well written and the motivation is very clear. It aims to solve two very important challenges in text-to-music generation: inference overhead and bad quality-diversity tradeoff. The proposed method makes a concrete understanding towards these challenges.

The methods including CFG distillation, diversity maximization, and the use of model weight interpolation are straighforward and simple enough that these ideas could be potentially easily and widely adapted to the deep generative model community.

The experimental results show concrete improvements on the quality-diversity tradeoff compared to the base model with or without CFG.

**Weaknesses:**

The major weakness of the CFG distillation is that it destroys the student model's ability to take different CFG scale as input, which is key in many downstream applications. I understand that this could be useful in applications where the inference strategy (i.e. CFG) is fixed by nature, but in many cases we need the flexibility to take different CFG scales at inference time. Therefore, I would expect the student model to take the CFG $\gamma$ as inputs as well, which could be done in a similar way how diffusion models are conditioned on timesteps.

The second weakness is that in Figure 5, the proposed model does not outperform CFG significantly. The quality is about the same, and the diversity is 57-vs-43. In addition, the diversity is much worse than the base model after model weight merging, indicating that the merged model still suffers from lack of diversity.

The third weakness is that the paper does not evaluate the proposed distillation method on other baseline models and tasks (e.g. audio generation and speech generation) -- especially open-sourced models -- which limits the contribution of the paper. As a result, it is unclear how general the proposed method is when applied in other data and tasks.

Minor weaknesses:

One minor weakness is that the paper does not analyze in depth the diversity drop of the vanilla CFG distillation. From Fig 2 one could infer it is most likely because the student model learns the CFG-modified logits well, but it is also possible that the KL loss in eq (2) amplifies it. It would be useful to study different divergence losses and different model capacities.

Another minor weakness is that Fig 3 is almost impossible to read. Please improve the display of this figure.

**Questions:**

Please refer to weaknesses.

---

> ### Author Response · Authors · 2024-11-22
>
> Thanks for your review and for saying that our paper “is extremely well written and the motivation is very clear. “
>
> > "The major weakness of the CFG distillation is that it destroys the student model's ability to take different CFG scale as input, which is key in many downstream applications. I understand that this could be useful in applications where the inference strategy (i.e. CFG) is fixed by nature, but in many cases we need the flexibility to take different CFG scales at inference time. Therefore, I would expect the student model to take the CFG as inputs as well, which could be done in a similar way how diffusion models are conditioned on timesteps."
> - Figure 1 shows that the tradeoff obtained with our approach outperforms the one obtained with varying the CFG coefficient.
> - For applications where retaining the CFG coefficient is crucial our approach can easily be extended with methods like CLP [1]. It would only require to swap the underlying RL algorithm with the CLP algorithm.
>
> > "The second weakness is that in Figure 5, the proposed model does not outperform CFG significantly. The quality is about the same, and the diversity is 57-vs-43."
> - LERP(0, 15) performs on par with CFG but outperforms CFG in terms of diversity which shows that we improved the diversity while preserving the quality. The statistical tests that we ran on the results of human evaluation shows that the results are significant which means that the human raters can really feel the difference in diversity.
>
> > "The third weakness is that the paper does not evaluate the proposed distillation method on other baseline models and tasks (e.g. audio generation and speech generation) -- especially open-sourced models -- which limits the contribution of the paper. As a result, it is unclear how general the proposed method is when applied in other data and tasks."
> - Indeed, we focused on music generation as it is an application where the importance of the quality-diversity tradeoff is clear as the diversity for creative tasks is important. Focusing on this task allowed us to run extensive human evaluations. Our proposed method does not have inherent limitations to other tasks and we leave the study in other applications to following work.
>
>
> [1] K. Wang et al. “Conditioned language policy: A general framework for steerable multiobjective finetuning”. In: arXiv preprint arXiv:2407.15762 (2024).

---

> > ### Comment · Reviewer_suju · 2024-11-26
> > **Reply from reviewer**
> >
> > Thank you very much for your response. I'm very happy to hear that you could apply methods like CLP to retain CFG generation. I highly recommend adding the discussion into the paper.
> >
> > I read review from other reviewers and think that the Frechet Distance with OpenL3 proposed by Stable Audio could be a good evaluation metric for your case, and I recommend running that evaluation.
> >
> > I will keep my score of 6 as I'm still on the positive side of this paper: the motivation, writing, and music generation results are good. I cannot give a higher score because the weaknesses, especially the 1st and 3rd point, still exist there.

---

> ### Author Response · Authors · 2024-11-26
>
> Thank you for your reply.
>
> > “I'm very happy to hear that you could apply methods like CLP to retain CFG generation. I highly recommend adding the discussion into the paper.”
>
> Thanks for your suggestion, we added this discussion in the paper.
>
> We agree with the importance of adding a metric used by the academic community. We added in Appendix G the FAD Score on MusicCaps with the CLAP and the VGGish backbones and the FAD on SongDescriber with CLAP. We see that without diversity (i.e. $\beta=0$) CFG Distillation matches the performance of CFG. Then, as we increase diversity, FAD decreases consistently.

---

> > ### Comment · Reviewer_suju · 2024-12-03
> > **Discussion**
> >
> > It is known that FAD isn't consistent with other metrics especially when it comes to music evaluation. I would recommend running the FD-OpenL3 which is more recent and could measure up to 48kHz music quality:
> >
> > https://github.com/Stability-AI/stable-audio-metrics/blob/main/examples/musiccaps_openl3_fd.py

---

### Official Review · Reviewer_5uP1 · 2024-11-01

**Soundness:** 3
**Presentation:** 3
**Contribution:** 3
**Rating:** 6
**Confidence:** 4

**Summary:**

In this work, the authors introduce a novel finetuning recipe for increasing the runtime efficiency of the model and improving diversity at the same time, through Diversity-rewarded CFG distillation and model merging. The authors show that by distilling the CFG behavior into the model with an RL-based diversity regularizer and then merging high diversity with high-quality model finetunes, they can achieve strong diversity and quality tradeoffs and improved results over the base model.

**Strengths:**

- Proposed CFG distillation method seems strong and is reasonably novel in the context of autoregressive LM models
- As a whole, proposed method is exceedingly clear, and while simple, shows the strengths of each facet of their 3-pronged method articulately.
- The significant time dedicated to discussing results and ablations is particular welcome, and the authors do a great job of exploring the design space of their method.
- The insight from Fig. 4 is in particular quite fascinating, do you have any idea why this merging behavior actually improves quality beyond the interpolating models?

**Weaknesses:**

Overall, while I think this is a reasonably strong work, there are a number of concerns I have addressed below that shift my overall recommendation to reject. In fixing some or all of these issues (in particular improving the discussion of CFG distillation novelty, using publicly available eval metrics such as FAD/CLAP/Recall/Coverage, including human eval error bars / statistical tests, and focusing more on the diversity model and its calibration in the main text) I would raise my score accordingly.

# Overclaiming of CFG Distillation Novelty:
In general, while the core contribution of the CFG distillation approach is interesting, I think it’s stated novelty is an over claim (such as the line 492-493 “In contrast, we are the first to distill CFG…for music,” which is false). While CFG Distillation may be novel in the case of autoregressive audio generation (MusicLM, MusicGen), CFG distillation is a common practice in diffusion models for the same reasons motivated in this paper (namely, reducing inference cost), normally coupled with other distillation techniques. While the authors rightly mention Meng et al., this work is rather old and the stated novelty of their “two-stage” approach vs. the authors’ “single-stage” approach (461-462) does not really hold, as most works after Meng et al. (and before the present work) use a single stage approach (and in general, the paper is presented as if the whole of CFG distillation is a fully novel contribution). Single-stage CFG distillation has been a standard practice in image domain distillation [1, 2], but more saliently has appeared in a number of audio (including music) generation [3,4] and music-specific generation [5] distillation works. As an actionable change, it would be useful to explicitly mention all of these connections in the paper (in both the introduction, the related works, and the section on CFG distillation), and make clear that the novelty of the proposed method comes from how they achieve CFG distillation (and the first to do so on autoregressive LM-based music models), rather than the *idea* of CFG distillation for music generation.


# Reduced Discussion on Diversity Model:
- It is odd that perhaps the most novel portion of the paper of the RL Diversity Objective (as both CFG distillation and model merging are standard practices) is relegated to most details of the core embedding model being in the appendix. To me, this is a core part of the paper (as defining diversity through some model underpins the entire method, and is very interesting), and it would be useful to include this in the main body of work (and I think there is a good deal of text in the results section that is redundant and could be condensed).

# Issues with Evaluation Suite:
- While the authors make no mention of open sourcing model weights or code (which in my opinion is fine and shouldn’t be counted as a negative against them), the overall evaluation suite **also** uses fully closed-source models, severely limiting the reproducibility of the work and how further research could use it as a baseline. While I am sure that the MuLan embeddings and User Preference score are fine metrics in their own right, they are wholly inaccessible to the wider community. To accurately assess how the proposed method performs, not only to their own baselines but also external models, it is imperative to include metrics that are publicly usable in the literature, such as CLAP score to assess text-relevance, FAD/MMD (with CLAP backbone) to assess audio quality w.r.t. a reference set (like SongDescriber), and so on. I very much recommend the authors include some number of these (such as FAD and CLAP) in order to contextualize their performance in the larger space of TTM generation. In parallel, running existing open source models (such as AudioLDM2 or Stable Audio Open) on the present closed source eval suite would also work.
- The definition of “quality score” (259-260) is somewhat divorced from the literature on TTM and generative evaluation. This seems to combine MuLan score (theoretically a measure of “text relevance”) and user preference, but this explicitly entangles these two different axes of behavior that are generally kept separate (as how previous works do not combine CLAP and FAD for instance [3,4]). It should either be explicitly stated as to *why* this combination can be made, or more explicitly simply uncombine them and present all results for user preference and MuLan score separately, as currently it is impossible to tell *which* of these metrics the proposed method actually improves.
- I have strong worries on the overall statistical power of the human eval. As the human eval is a core part of the presented results, both in order to prove the efficacy of the proposed method and to back-verify that the diversity model correlates with human perception, it is unclear what sort of statistical significance actually occurs here, as there seem to be no hypothesis tests run nor error bars reported for any of the human eval metrics, exacerbated by the fact that there seems to be only 3 users in the listening study (which is far lower than even the small sample sizes common in generative media research). Specifically, it would be useful to see paired t-tests run between each method on the scores (averaged across human raters) to determine significance.
- There is some circularity to the evaluation of diversity in this work, despite it being a core focus. Though the authors rightly identify that the quality score is a good measure as it is not used in training, using the diversity model for eval and RL fine-tuning by nature biases results towards the proposed method (as it directly optimizes for this result). This ignores the fact that there are a number of established different metrics for assessing diversity in generative media, including Inception Score (IS), Recall, Coverage, and Vendi Score to name a few. Including some or any one of these scores would be useful both to bring it more inline with existing TTM literature as well as assess the output diversity of the model in an unbiased fashion and help verify whether the trained diversity model correlates with these values.
- Along with the above 2 points, the overall justification of the diversity model’s success hinges on Figure 7. (Middle)’s analysis that the fine-tuned model shows similar Elo ranking to the base model. I’m not sure if Elo score is the best measure for this, as it is a purely relative measure and details on its exact calculation are not provided.  It would be better to show the *absolute* scores for each model (with error bars) and see how well this correlates with the diversity model, and in general justify more in the paper that the diversity model is reasonably calibrated given its underpinning for the whole work.

[1] Luo, Simian et al. “Latent Consistency Models: Synthesizing High-Resolution Images with Few-Step Inference.” ArXiv abs/2310.04378 (2023): n. pag.

[2] Yin, Tianwei et al. “Improved Distribution Matching Distillation for Fast Image Synthesis.” ArXiv abs/2405.14867 (2024): n. pag.

[3] Bai, Yatong et al. “ConsistencyTTA: Accelerating Diffusion-Based Text-to-Audio Generation with Consistency Distillation.” Interspeech 2024 (2023): n. pag.

[4] Saito, Koichi et al. “SoundCTM: Uniting Score-based and Consistency Models for Text-to-Sound Generation.” ArXiv abs/2405.18503 (2024): n. pag.

[5] Novack, Zachary et al. “DITTO-2: Distilled Diffusion Inference-Time T-Optimization for Music Generation.” ArXiv abs/2405.20289 (2024): n. pag.

**Questions:**

- Line (254-255): In discussing batch size, is this for the number of samples or the number of pairs samples generated? In general, it is not clear when discussing the RL objective how these pairs are chosen in the context of training, and whether the RL objective is only calculated on single pairs of samples or multiple pairwise comparisons.
- In Figure 3, is there a reason there is only a single LERP data point for the 4-way merge? It would be useful to see if further interpolation between the 4 models is able to achieve a strictly dominant Pareto front above the LERP(0,15) model, as the claim that “This validates that we can merge an arbitrary number of models.” (Line 359) seems somewhat under-validated in this context.
- Line 962, what is 1-similarity?
- While certainly not important, I wonder if the authors have insight as to how techniques from LLMs in the contrastive decoding literature (especially works like DoLa which are self-contrastive) are related to CFG for autoregressive LM TTM models. In fact, the update equations from CFG and contrastive decoding/DoLa are practically identical, as they both involve the log difference in probabilities between the base model and some “worse” reference model.

---

> ### Author Response · Authors · 2024-11-22
> **Response 1/2**
>
> We would like to thank Reviewer 5up1 for their review and try to address the expressed concerns.
>
> > "The insight from Fig. 4 is in particular quite fascinating, do you have any idea why this merging behavior actually improves quality beyond the interpolating models?"
> - In addition to interpolating between different behaviours, weight averaging also reduces variance [5,6] and memorisation [7] which leads to better generalization abilities, thus sometimes improved performances.
>
> > “exacerbated by the fact that there seems to be only 3 users in the listening study (which is far lower than even the small sample sizes common in generative media research)”
> - In our human evaluation, each couple (prompt, generation(s)) was indeed rated by 3 different raters. On the other hand, the total number of raters is 5 for the quality study and 6 for the diversity study. For the quality evaluation, each side-by-side comparison received 303 ratings (101 prompts x 3 ratings per prompt) while for the diversity evaluation each side-by-side comparison received 150 ratings (50 prompts x 3 ratings per prompts).
> - We can also note that for the selection of the raters, we used the same protocol as in Cideron et al: “We select raters for their experience listening to varied musical styles (>6 years) and fluency in written English.”
> - Our results are statistically significant as shown by the results of a t-test and a Wilcoxon test that we ran on all of our evals. The results are added to the revised paper in Appendix D.
>
>
> > “I’m not sure if Elo score is the best measure for this, as it is a purely relative measure and details on its exact calculation are not provided.”
> - The ELO score is a standard way to compute a ranking for different agents/models. We use the ELO score for several reasons: (1) the instructions of the raters are to explicitly provide a signal on which is the better model. Having to provide ratings improves rater signal quality, as demonstrated by prior human evaluations. (2) Using an ELO score avoids the calibration problem (i.e. raters have different scales when they rate answers), as we just consider an ordering of the answers independently of their values.
> - We use the classical way of calculating the ELO, as described on the wikipedia page, and used by the FIDE (International Chess Federation) https://en.wikipedia.org/wiki/Elo_rating_system#Mathematical_details . In particular, here is how we implement it. Let’s consider the win matrix W where W(i, j) is the number of wins of model i vs. model j, and D is the draw (symmetric) matrix where D(i, j) is the number of draws of model i vs. model j. For example, one win corresponds to the generation of model i being better than the one of model j (as rated by one rater). Then, we compute the gammas from these matrices: we use an iterative refinement algorithm with a loop that stops when the gammas have converged sufficiently. Inside this loop, the code updates each player's gamma based on their win/loss ratios against all other players. Specifically, for each player, it calculates a total_ratios value by summing the matches played against each opponent divided by the sum of the player's gamma and the opponent's gamma. The updated gamma for the player is then calculated as the player's total wins divided by this total_ratios value.
> After updating all gammas, we normalize to ensure the gammas sum to 1. Finally, we compute the elo with the following formula: elo = 400.0 * log10(gamma).
>
> > “It is odd that perhaps the most novel portion of the paper of the RL Diversity Objective (as both CFG distillation and model merging are standard practices) is relegated to most details of the core embedding model being in the appendix.”
> - We moved the details of the diversity measure in the main paper in section 2.2 (highlighted in blue).
>
> > “While CFG Distillation may be novel in the case of autoregressive audio generation (MusicLM, MusicGen), CFG distillation is a common practice in diffusion models for the same reasons motivated in this paper”
> - Thanks for the additional references. We updated the paper to make it clear that our approach is novel only in the case of LLMs. The updated text is highlighted in blue in the introduction, method and related work.
>
> > “Line (254-255): In discussing batch size, is this for the number of samples or the number of pairs samples generated? In general, it is not clear when discussing the RL objective how these pairs are chosen in the context of training, and whether the RL objective is only calculated on single pairs of samples or multiple pairwise comparisons.”
> - We use 2 as the number of generations per prompts. Hence, for a batch with a batch size = 128, there are 64 different prompts, and 2 generations per prompts. Each generation is sampled by the model. The diversity reward assigns a common score for the two generations coming from the same prompt.

---

> ### Author Response · Authors · 2024-11-22
> **Response 2/2**
>
> > “There is some circularity to the evaluation of diversity in this work, despite it being a core focus.”
> - We ran a robust human evaluation to explicitly break this circularity. Our experiments show that our diversity metric correlates with the diversity perceived by humans.
>
> > “I very much recommend the authors include some number of these (such as FAD and CLAP) in order to contextualize their performance in the larger space of TTM generation.”
> - We computed the CLAP scores for the set of prompts used for human evaluation (these prompts are the ones in the provided website) and corresponding generations.
> The results are the following:
>
>   $\beta=0$: 0.52 +/- 0.11
>
>   $\beta=15$: 0.52 +/- 0.11
>
>   LERP(0, 15): 0.52 +/- 0.11
>
>   base: 0.53 +/- 0.11
>
>   CFG: 0.53 +/- 0.11
>
> These results show that the CLAP score is not discriminative. We decided not to include it in the revision of the paper for this reason.
>
> - About the FAD score:
>   - FAD with MusicCaps has been used in several papers [1,2,3] but it was pointed out that FAD scores can be misleading. From [2]: “It is possible that due to those noisy samples, improvements in the quality of the generated audio might deteriorate the FAD on MusicCaps once a certain quality threshold is achieved.”
>   - More generally, the FAD score has been seen as a non reliable metric. According to [4]:“Despite being commonly used, prior work suggests that existing objective quality metrics including FAD fail to reliably predict the perceptual quality of generative music”.
>   - Hence, we decided not to include it in the paper and focus on reference-free metrics that were learned on human preference data.
>
> > “In Figure 3, is there a reason there is only a single LERP data point for the 4-way merge?”
> - Yes, it is a 4-way merge is an uniform averaging across all 4 checkpoints. Plotting a line is not straightforward as we would need to change 4 dimensions simultaneously.
>
> > “as the claim that “This validates that we can merge an arbitrary number of models.” (Line 359) seems somewhat under-validated in this context.”
> - We rephrased the sentence to soften the message (validate->suggest). As a side note, additional experiments made since time of submission confirm that we can merge (at least) up to 10 models, with gains growing as we merge more models.
>
> > “Line 962, what is 1-similarity?”
>  - Thanks for catching this error. This line was removed.
>
> > “While certainly not important, I wonder if the authors have insight as to how techniques from LLMs in the contrastive decoding literature (especially works like DoLa which are self-contrastive) are related to CFG for autoregressive LM TTM models. In fact, the update equations from CFG and contrastive decoding/DoLa are practically identical, as they both involve the log difference in probabilities between the base model and some “worse” reference model.”
> - The connections are indeed interesting! While the self supervised learning literature aimed at learning representations through data augmentation to which the model should be invariant, our CFG distillation actually benefits from an improved augmentation (CFG).
>
>
> [1] A. Agostinelli, T. I. Denk, Z. Borsos, J. Engel, M. Verzetti, A. Caillon, Q. Huang, A. Jansen, A. Roberts, M. Tagliasacchi, M. Sharifi, N. Zeghidour, and C. Frank. Musiclm: Generating music from text, 2023.
>
> [2] J. Copet, F. Kreuk, I. Gat, T. Remez, D. Kant, G. Synnaeve, Y. Adi, and A. Défossez. Simple and controllable music generation, 2023.
>
> [3] Novack, Zachary et al. “DITTO-2: Distilled Diffusion Inference-Time T-Optimization for Music Generation.” ArXiv abs/2405.20289 (2024).
>
> [4] Gui, A., Gamper, H., Braun, S., Emmanouilidou, D.: Adapting frechet audio distance for generative music evaluation. arXiv preprint arXiv:2311.01616 (2023)
>
> [5] Mitchell Wortsman, Gabriel Ilharco, Samir Ya Gadre, Rebecca Roelofs, Raphael Gontijo-Lopes, Ari S Morcos, Hongseok Namkoong, Ali Farhadi, Yair Carmon, Simon Kornblith, et al. Model soups: averaging weights of multiple fine-tuned models improves accuracy without increasing inference time. In ICML 2022.
>
> [6] A. Ramé, M. Kirchmeyer, T. Rahier, A. Rakotomamonjy, P. Gallinari, and M. Cord. Diverse weight averaging for out-of-distribution generalization. ICML, 2023.
>
> [7] Alexandre Ramé, Nino Vieillard, Léonard Hussenot, Robert Dadashi, Geoffrey Cideron, Olivier Bachem, and Johan Ferret. Warm: On the benefits of weight averaged reward models. ICML, 2024.

---

> > ### Comment · Reviewer_5uP1 · 2024-11-23
> >
> > Thank you to the authors for addressing most of my comments, and I have increased my score accordingly given the updates to the paper (in particular, the inclusion of statistical tests added references, and additional diversity analysis are very much appreciated). My overall score still currently leans towards rejection for two main reasons:
> > - I still wonder about the issue of combining MuLan and the user preference score into one single metric (as brought up in my initial review and not addressed in the rebuttal), as it is hard to tell what aspect of this score the proposed method is directly improving. In particular, given the reported CLAP results mentioned in the rebuttal, it would be very useful to see whether the improvements in quality are driven by user preference score alone (and thus that the proposed method has little effect on text adherence) or if MuLan score also improves (suggesting that it is simply CLAP that is an insensitive measure of text adherence).
> > - I disagree with the author's framing both of their CLAP results and the decision to not include FAD results. Re:CLAP, while the results seem nonsensitive, I still think it is interesting to report, as it either reflects an issue with CLAP or that the proposed method does not impact text adherence (which may strengthen the proposed method). For FAD, while one can certainly agree that MusicCaps is not a perfect reference dataset and FAD is not a perfect metric, it is still the academic standard used in almost all generative audio/music works, and is not unique to only using MusicCaps as a reference (in particular, a number of recent works use the much improved SongDescriber dataset, and Gui et al. note that while VGGish may be a bad backbone for FAD, using something like CLAP or Encodec is much more correlated with human perception). Additionally, it is not clear whether the proposed metrics are any better that FAD, and even FAD results, however noisy they may be, would still provide a useful artifact to the wider academic community to encourage reproducibility and benchmarking.

---

> ### Author Response · Authors · 2024-11-26
>
> Thank you for your reply.
>
> > “it would be very useful to see whether the improvements in quality are driven by user preference score alone (and thus that the proposed method has little effect on text adherence) or if MuLan score also improves”
>
> We added more details (Appendix B and Appendix C) about the individual variations of the User Preference metric and the MuLan metric for the different models. These additional plots show that the CFG with negative prompting provides a large boost in the user preference score. Secondly, for models with a large $\beta$, the drop of quality does not come from a drop of the User Preference but rather a drop in the text adherence. It shows that our approach improves the diversity by generating music a bit further away from the typical generation of a prompt, but it does not impact the audio quality. These findings are aligned with our qualitative evaluations.
>
> > “I still think it is interesting to report, as it either reflects an issue with CLAP”
>
> We added the results with the CLAP score in Appendix F.
>
> > “For FAD, while one can certainly agree that MusicCaps is not a perfect reference dataset and FAD is not a perfect metric, it is still the academic standard used in almost all generative audio/music works, and is not unique to only using MusicCaps as a reference (in particular, a number of recent works use the much improved SongDescriber dataset, and Gui et al. note that while VGGish may be a bad backbone for FAD, using something like CLAP or Encodec is much more correlated with human perception).”
>
> We agree with the importance of adding a metric used by the academic community. We added in Appendix G the FAD Score on MusicCaps with the CLAP and the VGGish backbones and the FAD on SongDescriber with CLAP. We see that without diversity (i.e. $\beta=0$) CFG Distillation matches the performance of CFG. Then, as we increase diversity, FAD decreases consistently.

---

> > ### Comment · Reviewer_5uP1 · 2024-11-27
> >
> > I thank the authors for their attention to detail and added changes, which have now fully addressed my concerns and have thus shifted my score to acceptance. I think the paper has significantly improved during the review process and would be a welcome inclusion at ICLR.

---

### Official Review · Reviewer_G1MN · 2024-11-04

**Soundness:** 3
**Presentation:** 4
**Contribution:** 2
**Rating:** 6
**Confidence:** 4

**Summary:**

The paper proposes a method called Diversity-Rewarded Classifier-Free Guidance (CFG) Distillation for improving generative models, particularly in creative tasks like music generation. The authors introduces two training objectives:
1. Distillation: The method distills CFG’s quality improvements directly into the model weights, which eliminates the extra computational overhead during inference.
2.Reinforcement Learning (RL) for Diversity: A reinforcement learning objective rewards the model for producing diverse outputs, balancing the quality-diversity trade-off.
The approach focuses on a text-to-music generation task, where it successfully improves quality and diversity, validated by human evaluations.

**Strengths:**

1. This work provides a new strategy for managing the quality-diversity balance in generative models and has potential applications in other areas of creative content generation.
2. To achieve both diversity and quality, instead of doing direct multi-objective training which can downgrade the performance, the authors first distill model in terms of audio quality with classifier free guidance, which increase the quality while maintaining the capability of text understanding to music.
3. The analysis of quality-diversity tradeoff provides further insights to music generation models, where quality and diversity are actually two factors that need to be balanced.
4. Great demos and great results! Did not listen to all of them, in particular, I like LERP 50/50 of “Melodic Danceable Brazilian Music with Percussions” the most.

**Weaknesses:**

1. For music, the terms “quality” and “diversity” are not obvious in the paper. “Quality” can not only refer to audio quality, but can also refer to other aspects of music, like style consistency, the key consistency, rhythmic patterns, etc. I assume “quality” in this paper means the audio quality only. "Diversity" is specified in Appendix C.1, which I think it "Quality" and "Diversity" should be mentioned clearly in the introduction or methodology section.
2. In terms of evaluation of music, using MuLan score is great in terms of text coherence of generated music, while assuming “quality” means audio quality, it would be nice to include Fréchet Audio Distance (FAD) (Kilgour et al., 2019).
3. RL with a diversity reward objective maybe is newly applied to the realm of music, but this is not new in the RL world. I believe at least the work “Effective Diversity in Population Based Reinforcement Learning” (Parker-Holder et al., 2020) is worth mentioning and I did not see this work is cited in the current paper.  I would like to see some editing in the paper to change the wordings. If I miss this citation or relevant citations, please point out and correct me.

**Questions:**

1. Rewarding to Weakness 2, would you mind including FAD or other metrics to state the quality?
2. "Diversity" is conceptualized as "Are these two audio excerpts derived from the same audio recording or not?". While this diversity is essential to study and learn, since we are dealing with music, is that possible that we can have better conceptualization of "Diversity"? For example, "Given one reference audio recording, how diverge are these two audio excerpts in terms of genre comparing to the reference audio?" In short, I think the word "Diversity" can be more music related.
3. Same logic applies to "Quality". Audio quality can be applied to speech as well. Since this work only focuses on music, can "Quality" be more music focused? For example, the structure and key coherence in the short excerpt.
4. Again, RL with a diversity reward objective seems like a general objective used or inspired from other fields, like population-based RL. While the application to music is novel, I don't think this is novel enough to be highlighted as novelty. Would you mind defending your argument of novelty? Why this objective special to music (this objective can be applied to speech as well)?

---

> ### Author Response · Authors · 2024-11-22
> **Response 1/2**
>
> Thank you for your review. We are happy that you found that we provided a ‘great demo and great results’.
>
>
> >  "“Quality” can not only refer to audio quality, but can also refer to other aspects of music, like style consistency, the key consistency, rhythmic patterns, etc. I assume “quality” in this paper means the audio quality only. "
>
> - In our paper we wanted to be as general as possible for the definition of quality. We rely on the MuLan score which indicates the adherence to the text and the User Preference score introduced in [5], which is learned on 300k pairwise preferences of music generations to approximate the average human opinion score.
> - In [4] it was shown that the User Preference score correlates with the acoustic audio quality and the musicality. Hence, our general quality score is sensitive to three important properties defined in [5]: the adherence to the text, the acoustic quality, and the musicality.
> - The human evaluation results for quality show that our quality metric positively correlates with the human preferences which is ultimately the metric we care about.
> - We can also note that our approach is general enough so that it can be applied to other “quality” metrics with metrics that explicitly favors ‘key consistency’ or ‘rhythmic patterns’ (the User Preference may already implicitly favor these aspects as it learned to rate the preferences of humans).
>
> > ""Diversity" is specified in Appendix C.1, which I think it "Quality" and "Diversity" should be mentioned clearly in the introduction or methodology section."
>
> - We added more details about the diversity in the methodology section (highlighted in blue in the revised text).
>
> > "RL with a diversity reward objective maybe is newly applied to the realm of music, but this is not new in the RL world. I believe at least the work “Effective Diversity in Population Based Reinforcement Learning” (Parker-Holder et al., 2020) is worth mentioning and I did not see this work is cited in the current paper."
>
> - Our approach of diversity is novel as it aims at optimising the diversity of a **single** agent while “existing quality diversity algorithms (Lehman & Stanley, 2011; Mouret & Clune, 2015; Cully et al., 2015; Cideron et al., 2020; Ding et al., 2024) aim at finding a population of agents with both high-quality and diverse behaviors.” (citing line 531-533 of the related work). This is a fundamental difference with the population-based line of work where individual agents may not be diverse.
> - We added the reference mentioned to the related work. It expands our list of existing quality-diversity algorithms that aim at improving the diversity of behaviors within a population of agents.
>
> > "In terms of evaluation of music, using MuLan score is great in terms of text coherence of generated music, while assuming “quality” means audio quality, it would be nice to include Fréchet Audio Distance (FAD) (Kilgour et al., 2019)."
> - FAD with MusicCaps has been used in several papers [1,2,3] but it was pointed out that the results can be misleading. From [2]: “It is possible that due to those noisy samples, improvements in the quality of the generated audio might deteriorate the FAD on MusicCaps once a certain quality threshold is achieved.”
> - More generally, the FAD score has been seen as a non reliable metric. According to [4]:“Despite being commonly used, prior work suggests that existing objective quality metrics including FAD fail to reliably predict the perceptual quality of generative music”.
> - Hence, we decided not to include it in the paper and focus on reference-free metrics that were learned on human preference data.
>
> > ""Diversity" is conceptualized as "Are these two audio excerpts derived from the same audio recording or not?". While this diversity is essential to study and learn, since we are dealing with music, is that possible that we can have better conceptualization of "Diversity"? For example, "Given one reference audio recording, how diverge are these two audio excerpts in terms of genre comparing to the reference audio?" In short, I think the word "Diversity" can be more music related."
> - In general our focus was to provide a general concept and methodology to improve diversity that can be extended to other modalities as well. It is indeed possible to specify and maybe compute diversity in terms of finer-grained musical concepts, but we believe it is out of scope for this work. Such a study would require involving musicologists and more elaborate qualitative and quantitative evaluations, which we leave for future work.

---

> ### Author Response · Authors · 2024-11-22
> **Response 2/2**
>
> > "Same logic applies to "Quality". Audio quality can be applied to speech as well. Since this work only focuses on music, can "Quality" be more music focused? For example, the structure and key coherence in the short excerpt."
> - In general, defining and computing musicality is a very challenging task that even professional musicologists struggle with, so we rely on general human preferences to overcome that limitation. Our extensive human evaluation shows that the metrics that we used are indeed correlated with the human perception of quality or diversity.
>
> > "Why this objective special to music (this objective can be applied to speech as well)?"
> - One of the key contributions of our approach is that it is **not** limited to music. As outlined in line 246, “We explore those questions on text-to-music generation, a creative task where quality and diversity are important factors. Indeed, a single music prompt could map to many different valid generations.“ We leave the study of the application of similar techniques to other modalities to future work.
>
>
> [1] A. Agostinelli, T. I. Denk, Z. Borsos, J. Engel, M. Verzetti, A. Caillon, Q. Huang, A. Jansen, A. Roberts, M. Tagliasacchi, M. Sharifi, N. Zeghidour, and C. Frank. Musiclm: Generating music from text, 2023.
>
> [2] J. Copet, F. Kreuk, I. Gat, T. Remez, D. Kant, G. Synnaeve, Y. Adi, and A. Défossez. Simple and controllable music generation, 2023.
>
> [3] Novack, Zachary et al. “DITTO-2: Distilled Diffusion Inference-Time T-Optimization for Music Generation.” ArXiv abs/2405.20289 (2024).
>
> [4] Gui, A., Gamper, H., Braun, S., Emmanouilidou, D.: Adapting frechet audio distance for generative music evaluation. arXiv preprint arXiv:2311.01616 (2023)
>
> [5] G. Cideron, S. Girgin, M. Verzetti, D. Vincent, M. Kastelic, Z. Borsos, B. McWilliams, V. Ungureanu, O. Bachem, O. Pietquin, M. Geist, L. Hussenot, N. Zeghidour, and A. Agostinelli. MusicRL: Aligning music generation to human preferences. In ICML, 2024.

---

> > ### Comment · Reviewer_G1MN · 2024-11-23
> >
> > Thank you for your rebuttal, those addresses most of my concerns, hence I will maintain my score.
> > FAD is a metric with drawbacks, but besides MuLan score (text music conference) and human preference, it's better to have a metric dedicated to quality. If FAD is not used, some metrics comparing two distributions (one is ground truth with high quality audio files) is needed, for example, KL Divergence.

---

> ### Author Response · Authors · 2024-11-26
>
> Thanks you for your reply.
>
> > "it's better to have a metric dedicated to quality."
>
> We would like to emphasise that the User Preference metric is a measure of quality. The ablations in Section 6 of [1] show that the User Preference is sensible to the audio quality and the musicality.
>
> We agree with the importance of adding a metric used by the academic community. We added in Appendix G the FAD Score on MusicCaps with the CLAP and the VGGish backbones and the FAD on SongDescriber with CLAP. We see that without diversity (i.e. $\beta=0$) CFG Distillation matches the performance of CFG. Then, as we increase diversity, FAD decreases consistently.
>
> In addition, we added more details (Appendix B and Appendix C) about the individual variations of the User Preference metric and the MuLan metric for the different models.
>
> [1] G. Cideron, S. Girgin, M. Verzetti, D. Vincent, M. Kastelic, Z. Borsos, B. McWilliams, V. Ungureanu, O. Bachem, O. Pietquin, M. Geist, L. Hussenot, N. Zeghidour, and A. Agostinelli. MusicRL: Aligning music generation to human preferences. In ICML, 2024.

---

### Official Review · Reviewer_EZJJ · 2024-11-04

**Soundness:** 2
**Presentation:** 3
**Contribution:** 3
**Rating:** 6
**Confidence:** 3

**Summary:**

This paper introduces a fine-tuning method that enhances both quality and diversity in generative models, targeting CFG’s limitations in computational cost and diversity reduction. The authors propose a two-part training objective: distilling CFG’s quality-enhancing effects into model weights and using reinforcement learning to encourage diversity. They also introduce model merging, which allows flexibility in balancing quality and diversity at deployment. Experiments on MusicLM show improvements over CFG, with human evaluations confirming superior quality-diversity trade-offs.

**Strengths:**

Originality: The paper presents an innovative approach to overcoming CFG’s limitations, particularly by combining distillation with a diversity-promoting reinforcement learning reward. It proposes a metric of diversity based on pair-level dissimilarity.
Quality: The methodology is robust, with well-designed experiments that rigorously evaluate quality-diversity trade-offs. The use of human evaluations adds significant credibility, and the proposed model-merging strategy for adjustable quality-diversity control demonstrates strong, reliable results.
Clarity: The paper clearly articulates its objectives, methods, and findings, making complex topics accessible. Detailed diagrams and well-structured explanations approach and results easy to understand, enhancing the overall readability and impact.
Significance : By improving both quality and diversity without increasing inference cost, this work addresses an important limitation in generative models. While primarily applied to music generation, the approach has potential implications across other generative domains, offering a valuable contribution to fields where creativity and efficiency are essential.

**Weaknesses:**

About the diversity training
According to equation 3, D(\theta) dependents on the expected value of dissimilarity of y given x. How many ‘y’s  are sampled given an x ? Besides, this paper does not further investigate the impact of the sample number of an x on the diversity training .


The definition of diversity is based on pair-level dissimilarity. It may be better to use some metrics like gini or Kurtosis to measure the global diversity of the generated songs.

**Questions:**

About music embedding
Why not use the open-source music embedding models to facilitate the reproducibility of this work?

Are there any results of embedding evaluation like acc of linear probing that can be public for the music embedding model?

---

> ### Author Response · Authors · 2024-11-22
>
> Thanks for your review. We are glad that you found that our paper ‘presents an innovative approach’, with a ‘robust’ methodology ‘offering a valuable contribution to fields where creativity and efficiency are essential’.
>
> > “About the diversity training According to equation 3, $D(\theta)$ depends on the expected value of dissimilarity of y given x. How many ‘y’s are sampled given an x ? Besides, this paper does not further investigate the impact of the sample number of an x on the diversity training.”
>
> Throughout our experiments, we use N=2 generations (y) given a prompt x. In the appendix C.3, we added a study of the impact of N on quality/diversity (the new content is highlighted in blue). Its results show that using N=2 is best for diversity.
>
> > "About music embedding Why not use the open-source music embedding models to facilitate the reproducibility of this work?"
>
> We wanted to use a SoTA speech embedding to music, so we implemented the embeddings from [1] and reproduced their results. This approach has the particularity that it was explicitly trained to map together similar song snippets and far apart different song snippets which is what we needed to have a reliable diversity measure.
>
> [1] M. Futeral, A. Agostinelli, M. Tagliasacchi, N. Zeghidour, and E. Kharitonov. Mad speech: Measures of acoustic diversity of speech. arXiv preprint, 2024.

---

> > ### Comment · Reviewer_EZJJ · 2024-12-03
> >
> > Many thanks for your rebuttal, your comment addresses my concerns.

---

> ### Author Response · Authors · 2024-11-29
>
> We would like to thank the reviewer again for their review.
>
> > “Are there any results of embedding evaluation like acc of linear probing that can be public for the music embedding model?”
>
> We thank the reviewer for their suggestion. We extracted a validation dataset made of 10k music not used during training. We computed top-1 accuracy on this eval dataset, using L2 similarity i.e. we transformed each song into 4 seconds chunks, we then computed the distance between a chunk and chunks coming either from other songs or coming from the same song. We assigned a score of 1 is the closest chunk was the one from the same song and 0 otherwise. Our fully trained model achieved a top-1 accuracy of 98.9%.

---

### Author Response · Authors · 2024-11-22
**General response to all reviewers**

We thank the reviewers for their constructive feedback, and are happy that they found that our work ‘presents an innovative approach’, with a ‘robust’ methodology, ‘that these ideas could be potentially easily and widely adapted to the deep generative model community’ and that it is ‘extremely well written’.

Following your suggestions, we made several additions/modifications to the paper:

- We added an analysis of the number of generations per prompt used to compute the diversity reward (in Appendix C.3).
- We added the full side-by-side human evaluation results with two statistical tests that show that the results of our human evaluation are statistically significant (in Appendix D).
- We changed the wording to make it clear that the CFG distillation is novel in the context of LLMs and we discussed its connection to prior work in the diffusion area. All the modifications to the text are highlighted in blue.
- We added more details about the diversity metric in the main text (see Section 2.2).
- We added a comparison between two measures of diversity based on a similarity matrix: the Vendi Score and 1-mean. We computed this comparison on the set of prompts used for human evaluation (see Appendix C.4).

About the CLAP score:
- For the 101 prompts used for the quality human evaluation, we generated music clips and computed the CLAP score with the official, open-source CLAP implementation (https://huggingface.co/laion/clap-htsat-unfused).  We generated 8 music clips per prompts, and computed a mean CLAP score and a standard deviation.
- Results:

  Clap score for beta=0: 0.52 +/- 0.11

  Clap score for beta=15: 0.52 +/- 0.11

  Clap score for LERP(0, 15): 0.52 +/- 0.11

  Clap score for base: 0.53 +/- 0.11

  Clap score for CFG: 0.53 +/- 0.11

 - These results show that the CLAP score is not discriminative. We decided not to include it in the revision of the paper for this reason.


About the FAD score:
-  FAD with MusicCaps has been used in several papers [1,2,3] but it was pointed out that the results can be misleading. From [2]: “It is possible that due to those noisy samples, improvements in the quality of the generated audio might deteriorate the FAD on MusicCaps once a certain quality threshold is achieved.”
- More generally, the FAD score has been seen as a non reliable metric. According to [4]:“Despite being commonly used, prior work suggests that existing objective quality metrics including FAD fail to reliably predict the perceptual quality of generative music”.
- Hence, we decided not to include it in the paper and focus on reference-free metrics that were learned on human preference data.



[1] A. Agostinelli, T. I. Denk, Z. Borsos, J. Engel, M. Verzetti, A. Caillon, Q. Huang, A. Jansen, A. Roberts, M. Tagliasacchi, M. Sharifi, N. Zeghidour, and C. Frank. Musiclm: Generating music from text, 2023.

[2] J. Copet, F. Kreuk, I. Gat, T. Remez, D. Kant, G. Synnaeve, Y. Adi, and A. Défossez. Simple and controllable music generation, 2023.

[3] Novack, Zachary et al. “DITTO-2: Distilled Diffusion Inference-Time T-Optimization for Music Generation.” ArXiv abs/2405.20289 (2024).

[4] Gui, A., Gamper, H., Braun, S., Emmanouilidou, D.: Adapting frechet audio distance for generative music evaluation. arXiv preprint arXiv:2311.01616 (2023)

---

### Meta-Review · Area_Chair_R4XJ · 2024-12-18

**Metareview:**

This paper introduces a fine-tuning method that enhances both quality and diversity in generative models, targeting classifier-free guidance’s limitations in computational cost and diversity reduction.

All reviewers described originality, clear description, and promising results with demonstration as the strenghths with borderline acceptance scores. Although reviewers pointed out some issues such as metrics and unclear quality definition, they seem to be not major issues.

In particular, this work addresses a music generation task, which is relatively less prevalent research work.

Considering the contributions and topic diversity, AC recommends accepting this paper.

**Additional Comments On Reviewer Discussion:**

The initial scores were 5, 6, 3, and 6.

In particular, 5uP1 raised major concerns such as method novelty, lack of discussion, and evaluation results.

During the rebuttal, the authors addressed most of concerns successfully, so the final scores are all 6 as scores.

---

### Decision · Program_Chairs · 2025-01-22

Accept (Poster)